# ALIGN AND FILTER: IMPROVING PERFORMANCE IN ASYNCHRONOUS ON-POLICY RL

## ABSTRACT

Distributed training and increasing the gradient update frequency are practical strategies to accelerate learning and improve performance, but both exacerbate a central challenge: *policy lag*, which is the mismatch between the behavior policy generating data and the learning policy being updated. Policy lag can hinder the scaling of on-policy learning algorithms to larger problems. In this paper, we identify the sources of policy lag caused by distributed learning and high update frequency. We use the findings to propose *total Variation-based Advantage aligned Constrained policy Optimization (VACO)* as a practical approach to mitigate policy lag. We empirically validate our method and show that it offers better robustness to policy lag in classic robotics RL tasks and a modern RL for LLM math reasoning task.

## 1 INTRODUCTION

Distributed reinforcement learning (RL) provides highly optimized and fast training pipelines by updating policies and collecting data simultaneously by using decentralized compute systems (Langer et al., 2020). In this setup, there are multiple compute nodes separately generating the data which is then sent to a learner node for processing. For example, multiple robots can run a policy to perform manipulation or locomotion tasks locally and transmit their collected experience to a central compute node to improve the policy (Gu et al., 2017; Rudin et al., 2022). One of the most common difficulties in this setup is the repeated failures and delays in communication which causes the data distribution to differ from the learning policy distribution (Yuan & Mahmood, 2022). This data distribution mismatch violates a key assumption in on-policy deep RL algorithms which assumes the behavioral and the target policy are the same (Sutton et al., 1998).

On-policy methods operate by generating data using environment rollouts with the current policy, and making conservative updates to keep close to the data distribution (Schulman et al., 2015a). For example, Proximal Policy Optimization (PPO) and its variants quantify the closeness of the distributions using the KL divergence (Schulman et al., 2017; Wang et al., 2020). Hence, the policy is optimized to increase reward, while bounding the divergence from the behavior policy. This allows a policy to make more mini-batch updates on the data to improve the policy's performance, but it widens the gap between the data distribution and the most up-to-date policy. This policy gap can lead to severe performance degradation or even full policy collapse (Nikishin et al., 2022). This limitation has previously been defined as *policy lag* in the literature and has been known to hinder policy improvement (Babaeizadeh et al., 2016; Petrenko et al., 2020).

In this paper, we provide a categorization sources of policy lag: **backward policy lag**, which arises from the initial mismatch between the behavior and learning policies, and **forward policy lag**, which accumulates as gradient updates cause the learning policy to diverge from the data distribution. This forward lag creates a critical trade-off: while more updates per batch of data may improve performance, they also increase the risk of policy degradation. We demonstrate that Total Variation (TV) divergence provides an effective tool for quantifying both sources of policy lag.

We propose Total Variation-based Advantage-aligned Constrained policy Optimization (VACO) to address the forward and backward policy lag which inherently exists in asynchronous RL. VACO is based on two main ideas: **advantage realignment** and **TV divergence-based filtering**. We use advantage realignment to estimate the advantage function of the learning policy from off-policy data which allows us to mitigate the problem of backward policy lag. To control the forward policy lag

that can arise from the policy optimization process, we then use TV divergence to filter data points in each minibatch that would amplify the effect of forward policy lag. We find that our filtering strategy is effective in maintaining a certain threshold on the value of the expected TV divergence without the additional complexity for the choice of hyperparameters for constraint satisfaction.

To study the effect of backward policy lag in various on-policy RL algorithms, we present a simulated asynchronous framework (Howes et al., 2025) for RL. We use this framework to simulate highly parallelized environments in various MuJoCo (Todorov et al., 2012) robotic tasks. We find that VACO offers better robustness to backward policy lag compared with PPO (Schulman et al., 2017), one of the most commonly used on-policy algorithms. We then investigate the forward policy lag present in finetuning large language models (LLMs) with RL towards math reasoning. We apply VACO to GRPO (Shao et al., 2024), the standard method in RL for LLMs, and find strong improvements in robustness to forward lags that are commonly present in these setups. In summary, we first provide a theoretical analysis of policy lag in asynchronous RL, categorizing its distinct sources. Building on this analysis, we propose VACO, a policy optimization algorithm designed for improved robustness to both forward and backward lag. We then empirically validate our method in two carefully constructed setups: highly parallelized robotics tasks and reasoning for LLMs. In both scenarios, we clearly improve robustness to policy lag compared to strong, standard RL baselines.

## 2 RELATED WORK

Our work investigates issues in the learning dynamics of on-policy RL algorithms when using data that arrives asynchronously to the learner agent. One of the key assumptions of on-policy methods is that data is collected with the current policy, making it challenging to reuse data or learn from off-policy samples (Sutton et al., 1998). A standard mechanism to alleviate this issue consists of defining a trust region constraint to update policies conservatively, allowing multiple updates on a given batch of data (Schulman et al., 2015a). Other approximation approaches have been previously studied to mitigate the need for second-order optimizers when optimizing the policy (Schulman et al., 2015a; 2017; Wang et al., 2020; Xie et al., 2025; Achiam et al., 2017b; Vuong et al., 2019). Off-policy variations have been proposed to introduce an adaptive change in the importance of the trust region regularization terms in the objective function (Meng et al., 2023; Fakoor et al., 2020; Queeney et al., 2021). Our research builds on the the TV divergence metric previously proposed to analyze the changes in data distribution in on-policy learning (Schulman et al., 2015a; Achiam et al., 2017a).

Despite recent advances, on-policy methods typically require large data batches for stable optimization (Andrychowicz et al.; Singla et al., 2024). Distributed learning architectures address this by parallelizing data collection and policy updates (Liu et al., 2024; Czech, 2021; Langer et al., 2020). In these setups, a central policy either sends its weights to parallel actors for data collection (Nair et al., 2015; Babaeizadeh et al., 2016; Mnih et al., 2016) or actors update local policies and send gradients back to the central learner (Kyzy et al., 2025). A trade off with asynchronous process approaches is introducing a mismatch, often called policy lag, between the acting policies and the policy being updated (Espeholt et al., 2018; Babaeizadeh et al., 2016). Our method mitigates policy lag by selectively applying only those updates that align with the direction of the advantage function.

Furthermore, synchronous policy updates are often impractical in real-world scenarios that require simultaneous inference and learning. This challenge is prevalent in diverse domains including real-time robotics, on-device mobile applications, and the training of large-scale language models, where system complexity makes asynchronous learning a necessity (Yuan & Mahmood, 2022; Gu et al., 2017; Radac & Chirla, 2025). Our algorithm addresses this need by effectively leveraging asynchronous samples, making it a practical solution for these demanding, real-world use cases.

## 3 BACKGROUND

### 3.1 REINFORCEMENT LEARNING SETUP

Reinforcement learning (RL) problems are typically formalized as **Markov Decision Processes (MDPs)** (Puterman, 2014). An MDP is a tuple $(\mathcal{S}, \mathcal{A}, \mathcal{R}, \mathcal{P}, \mu, \gamma)$ consisting of a state space $\mathcal{S}$, an action space $\mathcal{A}$, a reward function $\mathcal{R}(\cdot)$, a state transition probability function $\mathcal{P}(\cdot)$, an initial state

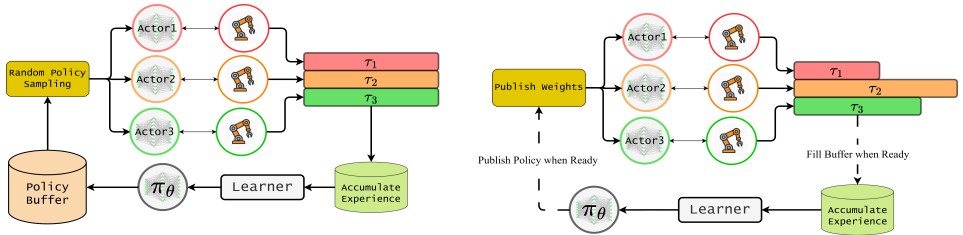

Mixture of Policies for Simulated Asynchronous RL        Real-World Asynchronous RL

Figure 1: **(left)** Simulated Asynchronous RL setup. After the end of each training phase we store the weights of the new policy in the policy buffer. Subsequently, we sample random policies from the buffer and assign them to the actors. We then generated the trajectories in a synchronous fashion. The setup aims to simulate an asynchronous RL training setup to control the backward policy lag. **(right)** The typical asynchronous RL training setup. The actors generate the trajectories and train the policy simultaneously. The actor would receive the new policy whenever it is ready. Therefore, the new dataset would contain trajectories from older policies.

distribution $\mu(\cdot)$, and a discount factor $\gamma \in [0, 1)$. The goal in RL problems is to optimize a policy $\pi : \mathcal{S} \times \mathcal{A} \to [0, 1]$ that maximize the expected discounted return of generated trajectories:

$$J(\pi) = \mathbb{E}_{s_0 \sim \mu, a_t \sim \pi(s_t), s_{t+1} \sim \mathcal{P}(s_t, a_t)} \left[ \sum_{t=0}^{\infty} \gamma^t r(s_t, a_t) \right].$$

Furthermore, the state-action value function is defined as the expected discounted return starting from a specified state-action pair,

$$Q_\pi(s, a) = \mathbb{E}_{a_t \sim \pi(s_t), s_{t+1} \sim \mathcal{P}(s_t, a_t)} \left[ \sum_{t=0}^{\infty} \gamma^t r(s_t, a_t) | s_0 = s, a_0 = a \right],$$

and can be used derive the state value function $V\pi(s) = \mathbb{E}_{a \sim \pi}[Q_\pi(s, a)]$ Finally, the advantage function is defined as $A_\pi(s, a) = Q_\pi(s, a) - V_\pi(s)$. When generating samples from the policy $\pi$, the expected value of the $A_\pi(s, a)$ is zero: $\mathbb{E}_{a \sim \pi(s)}[A_\pi(s, a)] = \mathbb{E}_{a \sim \pi(s)}[A_\pi(s, a)] = \mathbb{E}_{a \sim \pi(s)}[Q_\pi(s, a)] - V_\pi(s) = 0$.

### 3.2 ASYNCHRONOUS ON-POLICY RL

In a typical asynchronous RL training setup multiple independent agents are used to gather trajectories. In this setup, the agents update their policy with the most final version whenever it is ready. Each typical iteration of policy training involves two phases: trajectory generation and policy optimization. The trajectory generation phase gathers data until the replay buffer is filled. Next, the policy optimization phase updates the policy on the data for multiple epochs. To this end, consider iteration $T$, the trajectories are generated using a behavior policy $\beta_T$ that, in the asynchronous setup, can be specified as an episodic mixture distribution (Wiering & Van Hasselt (2008)):

$$\begin{aligned}
\beta_T(a|s) &= \mathbb{E}_{i \sim \mathcal{M}(0,...,T)} \left[ \pi_i(a|s) \right] \\
\Pr(s|\tau \sim \beta_T(\mu)) &= \mathbb{E}_{s_0 \sim \mu, i \sim \mathcal{M}(0,...,T)} [\Pr(s|\tau \sim \pi_i)]
\end{aligned} \tag{1}$$

where the mixture distribution $\mathcal{M}$ is usually unknown apriori and depends on the specific training setup. Due to the nature of the asynchronous setup, it can be observed that even if the policy is parameterized by a probability density function (e.g. a Gaussian distribution), the mixture distribution might not have the same parametrization. In the theoretical analysis that follows in this paper, $\beta_T$ will be used to denote the behavior at iteration $T$. Furthermore, the policy for the next iteration $T + 1$ is obtained by training the policy at iteration $T$ for multiple epochs:

$$\theta_{T+1}^{(n)} \leftarrow \theta_{T+1}^{(n-1)} + \eta \nabla_\theta L(\theta_{T+1}^{(n-1)}) \tag{2}$$

where $\theta_{T+1}^{(n)}$ denotes the policy parameter at epoch $n$, $\theta_{T+1}^{(0)} = \theta_T$, $\eta$ the learning rate and $L$ the loss function used to optimize the policy. The asynchronous RL setup is depicted in Figure 1.

### 3.3 TRUST REGION POLICY OPTIMIZATION

Trust region policy optimization (TRPO) (Schulman et al., 2015a) is one of the first successful attempts at developing on-policy RL methods with monotonic policy improvement guarantees. TRPO's policy improvement guantee is based on the performacne difference lemma:

**Lemma 1** *(Kakade & Langford, 2002) For any two policies $\pi$ and $\pi'$ and the initial state distribution $\mu$:*

$$J(\pi') - J(\pi) = \mathbb{E}_{\tau \sim \pi'|s_0 \sim \mu} \left[ \sum_{t=0}^{\infty} \gamma^t A_\pi(s_t, a_t) \right] = \frac{1}{1-\gamma} \mathbb{E}_{s \sim d^{\pi'}, a \sim \pi'(s)} [A_\pi(s, a)] \quad (3)$$

where $d^\pi(s) = (1 - \gamma) \sum_{t=0}^{\infty} \gamma^t \mathcal{P}(s_t = s|\mu, \pi)$ is the discounted future state distribution. We refer the readers for the proof of the performance difference lemma to Section B.1 or (Kakade & Langford, 2002; Schulman et al., 2015a; Agarwal et al., 2019). The performance difference equation can effectively relate two policies using the advantage function.

However, optimizing the policy using this equation can be challenging due to the dependency of $d^{\pi'}$ on $\pi'$. To this end, (Kakade & Langford, 2002; Schulman et al., 2015a) propose the approximate version of the performance difference as:

$$L_\pi(\pi') = \frac{1}{1-\gamma} \mathbb{E}_{s \sim d^\pi, a \sim \pi(s)} \left[ \frac{\pi'(a|s)}{\pi(a|s)} A_\pi(s, a) \right]. \quad (4)$$

The approximate version allows the optimization of the performance of the policy with a conservative step size since it matches the original version to the first order. The performance difference between two policies can then be characterized using Equation 4[1]:

**Theorem 1** *(Achiam et al., 2017a) For any two policies $\pi'$ and $\pi$, and defining $\epsilon^{\pi'} = \max_{s \in \mathcal{S}} |\mathbb{E}_{a \sim \pi'}[A_\pi(s, a)]|$, denoting $D_\pi^\pm(\pi') = \frac{L_\pi(\pi')}{1-\gamma} \pm \frac{2\gamma\epsilon^{\pi'}}{(1-\gamma)^2} \mathbb{E}_{s \sim d^\pi}[D_{TV}(\pi' \| \pi)[s]]$ where $D_{TV}(\pi' \| \pi)[s] = (1/2) \sum_{a \in \mathcal{A}} |\pi'(a|s) - \pi(a|s)|$, the performance difference can be bounded as:*

$$D_\pi^-(\pi') \leq J(\pi') - J(\pi) \leq D_\pi^+(\pi') \quad (5)$$

The bound in Equation 5 is tight since at $\pi' = \pi$ we get $D_\pi^\pm(\pi') = 0$. Hence, with iterative optimization of the lower bound $D_\pi^-(\pi')$, the policy enjoys monotonic improvement on the performance.

While the lower bound on the performance difference in Equation 5 is suitable for optimizing the policy, the exact optimization of the lower bound would be challenging since $\epsilon^{\pi'} = \max_{s \in \mathcal{S}} |\mathbb{E}_{a \sim \pi'}[A_\pi(s, a)]|$ is challenging to evaluate. The developers of TRPO derive the lower bound equation as $\frac{L_\pi(\pi')}{1-\gamma} - \frac{4\gamma\epsilon}{(1-\gamma)^2} \max_{s \in \mathcal{S}} D_{KL}(\pi, \pi')$ and proposed constraining the average KL divergence $\bar{D}_{KL}(\pi, \pi') = \mathbb{E}_{s \sim d^\pi}[D_{KL}(\pi(.|s), \pi'(.|s)] \leq \delta$ as a heuristic approximation for the max operator,

$$\theta^* = \text{argmax}_{\theta \in \Theta} \mathbb{E}_{s \sim d^{\pi_{\theta_{old}}}, a \sim \pi_{\theta_{old}}(s)} \left[ \frac{\pi_\theta(a|s)}{\pi_{\theta_{old}}(a|s)} A_{\pi_{\theta_{old}}}(s, a) \right]$$
$$s.t. \ \bar{D}_{KL}(\pi_{\theta_{old}}, \pi_\theta) \leq \delta, \quad (6)$$

where $\pi_{\theta_{old}}$ is the initial policy used to collect the trajectories. Therefore, with a conservative enough threshold $\delta$, the policy can have non-decreasing returns.

### 3.4 PROXIMAL POLICY OPTIMIZATION

The same idea of trust region policy optimization in Equation 6 can be applied to Equation 5 (Achiam et al., 2017a), replacing the constraint with the total variational difference

---

[1](Schulman et al., 2015a) also provide similar variations of the theorem. However, the version discussed in (Achiam et al., 2017a) uses the expected value of $D_{TV}$ which is more practical to use.

Figure 2: **Algorithmic Choices in VACO**. **(Left) VACO vs. PPO Clipping**: To maintain a fixed Total Variation (TV) divergence, VACO(shown as 'TV Filtering') selectively removes gradients that would increase the TV divergence. In contrast, PPO naively clips gradients if their policy ratio exceeds a predefined threshold. **(Right) IMPALA vs. Advantage Realignment**: While IMPALA estimates advantage values for the most recent policy using an asynchronously generated dataset, VACO's 'Advantage Realignment' first aligns the dataset to the initial policy of the optimization process, then iteratively optimizes based on this aligned dataset. This approach significantly reduces the computational load compared to IMPALA's on-the-fly realignment.

$\mathbb{E}_{s \sim d^{\pi_{\theta_{old}}}}[D_{TV}(\pi_{\theta_{old}} \| \pi_\theta)[s]] \leq \delta$. The $\mathbb{E}_{s \sim d^{\pi_{\theta_{old}}}}[D_{TV}(\pi_{\theta_{old}} \| \pi_\theta)[s]]$ can further be approximated by sampling from the rollout trajectories of $\pi_{\theta_{old}}$ as:

$$\mathbb{E}_{s \sim d^{\pi_{\theta_{old}}}}[D_{TV}(\pi_{\theta_{old}} \| \pi_\theta)[s]] = \frac{1}{2}\mathbb{E}_{s \sim d^{\pi_{\theta_{old}}}, a \sim \pi_{\theta_{old}}(s)}\left[\left|\frac{\pi_\theta(a|s)}{\pi_{\theta_{old}}(a|s)} - 1\right|\right] \quad (7)$$

To optimize the policy with the TV divergence, Proximal Policy Optimization (PPO) proposed the clipped surrogate objective loss:

$$L_{CLIP}(\theta) = \mathbb{E}_{s \sim d^{\pi_{\theta_{old}}}, a \sim \pi_{\theta_{old}}(s)}[\min\{r_\theta(s,a)A(s,a), clip(r_\theta(s,a), 1-\delta, 1+\delta)A(s,a)\}] \quad (8)$$

where $r_\theta(s,a) = \frac{\pi_\theta(a|s)}{\pi_{\theta_{old}}(a|s)}$. The clipping technique removes any gradients in the minibatch that have a ratio higher outside the $(1-\delta, 1+\delta)$ range. The clipping mechanism is designed as an effective heuristic to bound $D_{TV}$ with computationally efficient operations, and has been previously utilizes in other PPO variants Queeney et al. (2021).

# 4 METHODOLOGY

## 4.1 PPO PERFORMANCE WITH OFF-POLICY DATA

To understand why the approximate version of the performance difference in Equation 4 causes a performance degradation in PPO, we first provide the generalization of the performance difference lemma with respect to an off-policy distribution (in the following, general notation $\pi$ is used to denote any subsequent policy that might be obtained during the training process):

**Lemma 2** *Consider the policy at iteration $T$ as $\pi_T$ and the behavior policy $\beta_T$. Then, denoting $\epsilon^\pi = \max_{s \in \mathcal{S}} |\mathbb{E}_{a \sim \pi}[A_{\beta_T}(s,a)]|$, the performance difference between any policy $\pi$ and $\pi_T$ can be lower bounded using the state distribution of $\beta_T$ as:*

$$J(\pi) - J(\pi_T) \geq$$
$$\frac{1}{1-\gamma}\mathbb{E}_{s \sim d^{\beta_T}, a \sim \beta_T(s)}\left[\frac{\pi(a|s)}{\beta_T(a|s)}A_{\beta_T}(s,a) - \frac{\pi_T(a|s)}{\beta_T(a|s)}A_{\beta_T}(s,a)\right.$$
$$\left. - \left(\frac{2\gamma\epsilon^\pi}{1-\gamma}D_{TV}(\beta_T\|\pi)[s] + \frac{2\gamma\epsilon^{\pi_T}}{1-\gamma}D_{TV}(\beta_T\|\pi_T)[s]\right)\right]$$
$$(9)$$

Using the lower bound in Equation 9, the sources of policy lag can then be identified:

**Forward Policy Lag** As more gradient updates are made to the policy, the $\frac{2\gamma\epsilon^\pi}{1-\gamma}D_{TV}(\beta_T\|\pi)[s]$ penalty term on the lower bound becomes larger if it is not controlled. Hence, the risk of the penalty

term overcoming the advantage term $\frac{\pi(a|s)}{\beta_T(a|s)}A_{\beta_T}(s,a)$ increases. In fact, various on-policy tricks such as the clipping mechanism in PPO or the KL constraint in TRPO aim to address this.

**Backward Policy Lag** The mismatch between the initial learning policy $\pi_T$ and the behavior policy $\beta_T$ creates an inevitable penalty on the performance difference $\left(\frac{\pi_T(a|s)}{\beta_T(a|s)}A_{\beta_T}(s,a) + \frac{2\gamma\epsilon^{\pi_T}}{1-\gamma}D_{TV}(\beta_T\|\pi_T)[s]\right)$. Therefore, any initial distribution mismatch causes performance degradation.

To observe this point, consider the lower bound when $\pi = \pi_T$:

$$J(\pi_T) - J(\pi_T) = 0 \geq \frac{1}{1-\gamma}\mathbb{E}_{s\sim d^{\beta_T},a\sim\beta_T(s)}\left[-\frac{4\gamma\epsilon^{\pi_T}}{1-\gamma}D_{TV}(\beta_T\|\pi_T)[s]\right] \tag{10}$$

Any mismatch in distribution between $\beta_T$ and $\pi_T$ creates a strictly negative lower bound. For that reason, it may be challenging to optimize the policy such that the term $\frac{\pi(a|s)}{\beta_T(a|s)}A_{\beta_T}(s,a)$ would outperform the other three penalty terms and create a positive lower bound.

### 4.2 OFF-POLICY PERFORMANCE DIFFERENCE

In Lemma 2 we showed that the approximate version of the performance difference poses an unavoidable penalty. However, the lower bound can be improved.

**Lemma 3** *Consider the policy at iteration $T$ as $\pi_T$ and the behavior policy $\beta_T$. Then, denoting $\epsilon^\pi = \max_{s\in\mathcal{S}}|\mathbb{E}_{a\sim\pi}[A_{\pi_T}(s,a)]|$, the performance difference between any policy $\pi$ and $\pi_T$ can be lower bounded using the state distribution of $\beta_T$ as:*

$$J(\pi) - J(\pi_T) \geq \frac{1}{1-\gamma}\mathbb{E}_{s\sim d^{\beta_T},a\sim\beta_T(s)}\left[\frac{\pi(a|s)}{\beta_T(a|s)}A_{\pi_T}(s,a) - \frac{2\gamma\epsilon^\pi}{1-\gamma}D_{TV}(\beta_T\|\pi)[s]\right] \tag{11}$$

The main difference between Equation 5 and Equation 11 can be attributed to the advantage function. This difference provides an important effect. To see this (as mentioned in Section 3.1), notice that at $\pi = \pi_T$ we get $\mathbb{E}_{s\sim d^{\beta_T},a\sim\beta_T(s)}\left[\frac{\pi_T(a|s)}{\beta_T(a|s)}A_{\pi_T}(s,a)\right] = \mathbb{E}_{s\sim d^{\beta_T},a\sim\pi_T(s)}[A_{\pi_T}(s,a)] = 0$ and $\epsilon^{\pi_T} = \max_{s\in\mathcal{S}}|\mathbb{E}_{a\sim\pi_T}[A_{\pi_T}(s,a)]| = 0$. Therefore, while the forward policy lag still exists in the lower bound, the off-policy performance difference offers zero backward policy lag. Moreover, we discuss two main modifications to optimize the policy using Equation 11.

#### 4.2.1 ADVANTAGE REALIGNMENT

As discussed, the key difference between Equation 11 and Equation 5 is the use of the learning policy's advantage, $A_{\pi_T}$, versus the behavior policy's advantage, $A_{\beta_T}$. Since trajectories are generated by the behavior policy $\beta_T$, we can only directly estimate $A_{\beta_T}$. This creates an off-policy evaluation challenge: how to estimate the value of one policy using data generated by another (Munos et al., 2016; Espeholt et al., 2018).

A prominent solution, particularly in asynchronous settings, is the V-trace method from IMPALA (Espeholt et al., 2018). Given trajectories generated by a behavior policy $\beta$, the V-trace target for a policy $\pi$ at any state $s$ is defined as:

$$v_\pi(s) := V_\beta(s) + \sum_{t=0}^{\infty}\gamma^t\left(\prod_{i=0}^{t-1}c_i\right)\rho_t\left(r_t + \gamma V_\beta(s_{t+1}) - V_\beta(s_t)\right) \tag{12}$$

where $c_i = \min\left(\bar{c}, \frac{\pi(a_t|s_t)}{\beta(a_t|s_t)}\right)$, $\rho_t = \min\left(\bar{\rho}, \frac{\pi(a_t|s_t)}{\beta(a_t|s_t)}\right)$. Relating to the off-policy performance difference, we use Equation 12 to estimate the value function of $\pi_T$ using the trajectories sampled from $\beta_T$. Furthermore, similar to generalized advantage estimation (GAE) (Schulman et al., 2015b) method used by PPO, the V-trace target can also be efficiently computed recursively in a trajectory. The recursive rule and the advantage function are defined as:

$$v_\pi(s_t) = V_\beta(s_t) + \rho_t(r_t + \gamma V_\beta(s_{t+1}) - V_\beta(s_t)) + \gamma\lambda c_t(v_{t+1} - V(s_{t+1})) \tag{13}$$

$$A_{vtrace}(s_t, a_t) = r_t + \gamma v_\pi(s_{t+1}) - V_\beta(s_t) \tag{14}$$

where $\lambda \in [0, 1]$ is the discounting parameter similar to GAE which controls the bias-variance tradeoff of the estimations. We note that the subtraction of $V_\beta(s_t)$ is used for variance reduction.

As noted in (Espeholt et al., 2018), $\bar{\rho}$ controls the fixed point of the policy that the V-trace converges to. Hence, for $\bar{\rho} = \infty$ the target results in $V_\pi$. On the other hand, $\bar{c}$ controls the rate of convergence of the target through the contraction coefficient $\eta$.

The key difference between IMPALA and our Advantage Realignment method lies in the frequency of advantage estimation. IMPALA continuously re-estimates the advantage function for the current policy at each step, effectively treating the problem as a series of on-policy updates. In contrast, Advantage Realignment calculates the advantage function only once for the initial learning policy, $\pi_T$, and then iteratively optimizes Equation 5. This avoidance of repeated calculations makes Advantage Realignment significantly more computationally efficient than IMPALA. Other than architectural differences, Equation 11 allows the optimization process to be more robust to the off-policy correction errors in estimation compared with IMPALA since the target advantage function is fixed rather than the variability of the target in IMPALA.

### 4.2.2 FILTERING-BASED CONSTRAINED POLICY OPTIMIZATION

Beyond accurately estimating the advantage in Equation 11, it is also crucial to minimize the penalty term. However, as discussed in Section 3, directly controlling the max operator within the error term $\epsilon^\pi$ is challenging. Therefore, inspired by prior work in constrained policy optimization (Achiam et al., 2017a), we reframe the problem. We minimize our objective subject to a constraint on the Total Variation (TV) divergence, which in turn mitigates the impact of $\epsilon^{\pi\,2}$:

$$\theta^* = \text{argmax}_{\theta \in \Theta} \mathbb{E}_{s \sim d^{\beta_T}, a \sim \beta_T(s)} \left[ \frac{\pi(a|s)}{\beta_T(a|s)} A_{\pi_T}(s, a) - \frac{2\gamma\epsilon^\pi}{1-\gamma} D_{TV}(\beta_T \| \pi)[s] \right]$$

$$\Downarrow$$

$$\theta^* = \text{argmax}_{\theta \in \Theta} \mathbb{E}_{s \sim d^{\beta_T}, a \sim \pi_\theta(s)} [A_{\pi_T}(s, a)]$$

$$s.t. \ \mathbb{E}_{s \sim d^{\beta_T}} [D_{TV}(\beta_T \| \pi_\theta)[s]] \leq \delta/2$$

(15)

A common approach for constrained optimization is to convert the problem into an unconstrained one using the Lagrange method, which guarantees constraint satisfaction asymptotically (Altman, 2021). However, in practice, satisfying the constraint at each iteration is challenging due to the initial choice of the multiplier and the limited data per update. To better satisfy the constraint at each step, our approach directly analyzes the relationship between the loss gradient and the constraint gradient. Let $\theta$ be the policy parameters; we consider the gradient of the loss function and the gradient of $D_{TV}$ with respect to $\theta$:

$$\nabla_\theta \mathbb{E}_{s \sim d^{\beta_T}, a \sim \beta_T(s)} \left[ \frac{\pi_\theta(a|s)}{\beta_T(a|s)} A_{\pi_T}(s, a) \right] = \mathbb{E}_{s \sim d^{\beta_T}, a \sim \beta_T(s)} \left[ \frac{\pi_\theta(a|s)}{\beta_T(a|s)} \nabla \log \pi_\theta(a|s) A_{\pi_T}(s, a) \right]$$

(16)

$$\nabla_\theta \mathbb{E}_{s \sim d^{\beta_T}, a \sim \beta_T(s)} \left[ \left| \frac{\pi_\theta(a|s)}{\beta_T(a|s)} - 1 \right| \right] = \mathbb{E}_{s \sim d^{\beta_T}, a \sim \beta_T(s)} \left[ \frac{\pi_\theta(a|s)}{\beta_T(a|s)} \nabla \log \pi_\theta(a|s) sgn(\pi_\theta(a|s) - \beta_T(a|s)) \right]$$

(17)

From the comparison it can be observed that whether or not a specific data point can contribute to the increase or decrease of $D_{TV}$ can be determined from the comparison between the sign of $A_{\pi_T}(s, a)$ and $sgn(\pi_\theta(a|s) - \beta_T(a|s))$. Using this property, if $\mathbb{E}[D_{TV}]$ exceeds the threshold on that minibatch, we propose detaching the gradients of the data points from the minibatch that lead to increase of the TV divergence:

$$if \ \mathbb{E}_{s \sim d^{\beta_T}} [D_{TV}(\pi_\theta \| \beta_T)[s]] > \delta \Rightarrow remove \ i : A_{\pi_T}(s_i, a_i) sgn(\pi_\theta(a_i|s_i) - \beta_T(a_i|s_i)) > 0$$

(18)

Our proposed filtering method enables modifying the loss function such that the constraint is satisfied without additional hyperparameters to control. In addition, the modification is also compatible

---

[2] The threshold is specified as $\delta/2$ to match it with $1/2$ used in the definition of $D_{TV}$ and have $\delta$ reflect the same threshold that the clipping mechanism aims to address.

with the maximum entropy objective by sampling the actions from $\beta_T$ and using the importance sampling ratio:

$$\mathcal{H}(\pi_\theta) = -\mathbb{E}_{s \sim d^{\beta_T}, a \sim \beta_T(s)} \left[ \frac{\pi_\theta(a|s)}{\beta_T(a|s)} \log \pi_\theta(a|s) \right] \tag{19}$$

$$\nabla_\theta L_{\beta_T}(\theta) = \mathbb{E}_{s \sim d^{\beta_T}, a \sim \beta_T(s)} \left[ \frac{\pi_\theta(a|s)}{\beta_T(a|s)} \nabla_\theta \log \pi_\theta(a|s)(A_{\pi_T}(s,a) - c_\mathcal{H} \log \pi_\theta(a|s)) \right] \tag{20}$$

where $c_\mathcal{H}$ is the max entropy coefficient. The filtering technique allows to conservatively improve the performance of the policy and maximize the entropy simultaneously without the risk of violating the TV constraint.

Intuitively, the TV-filtering method operates as a controller. When the TV divergence value between the main policy and the behavior policy is lower than the threshold, the method allows for the use of all the data points. On the other hand, when the divergence value is above the threshold, it uses the data points that would lead to the decrease of the divergence value.

Overall, we call the combined ideas of Advantage Realignment and TV-based Filtering, VACO. The main differences between Realignment and IMPALA and TV-based Filtering and PPO are depicted in Figure 2 and the Pseudocode of the algorithm can be found in Algorithm 1.

## 5 EXPERIMENTS

We validate our method experimentally by examining the effect of policy lag on the performance of standard RL algorithms. We then implement VACO to address 1. backward policy lag due to an asynchronous MuJoCo setup and 2. forward policy lag due to an RL for LLM reasoning setup.

### 5.1 BACKWARD POLICY LAG IN MuJoCo

As mentioned in the previous section, the main source of policy lag can be attributed to the mismatch between the current policy and the behavior policy. The mismatch can initially be presented in terms of the behavior policy in the state distribution and further amplified through consecutive gradient descent passes on the loss function. The former happens in asynchronous/distributed sampling setups and the latter in each gradient descent pass. While various asynchronous RL setups have been proposed in the past, obtaining reproducible results is a challenging task due to the underlying non-determinism on the hardware level (Huang et al., 2023). To this end, in order to study policy lag in a controlled environment, we propose to use a simulated asynchronous setup (Howes et al., 2025). The difference between the simulated platform and the real-world setup is highlighted in Figure 1. As discussed in Section 3.2, the asynchronous setup can be described as a mixture of policies setup with an unknown mixing rule. Hence, we aim to replicate the process in a controllable setting. In our simulated setup we use a policy buffer of a fixed capacity to save the past policies. After the end of the training phase we save the weights of the policy in the buffer and randomly sample from the buffer and assign them to the actors. Hence, we sample data synchronously to generate a dataset of equal sized trajectories. This setup allows the study of the mismatch between the behavior policy and the current policy using a mixture of policies setup. After the training phase, the final policy is rolled out in the environment for multiple episodes and the return of the policy is calculated by taking the average over the return of the trajectories.

We implement VACO in the CleanRL codebase (Huang et al., 2022) on top of PPO. From Pinsker's inequality we have $D_{TV}(p\|q)^2 \leq D_{KL}(p\|q)$. Hence, since it has become the standard trick (Schulman et al., 2017; Ziegler et al., 2019; Stiennon et al., 2020) to minimize policy lag for PPO, we use KL divergence penalty with various coefficient values to run PPO and report the runs with the best performing return. We indicate "PPO-KL Penalty=0" as the vanilla version of PPO (known as PPO-Clip) and "PPO-KL Penalty=1" as the one with KL-regularization with coefficient of one. Furthermore, we use Simple Policy Optimization (SPO) (Xie et al., 2025) which uses the square of the TV divergence $\mathbb{E}[(\frac{\pi}{\beta} - 1)^2]$ as a penalty term and removes the clipping mechanism. Additionally, we implement IMPALA Espeholt et al. (2018) using the cleanRL baseline to evaluate the method that continuously re-estimates the advantage function of the new policy. Regarding the choice of hyperparameters, we use the standard default hyperparameters offered in the CleanRL codebase and

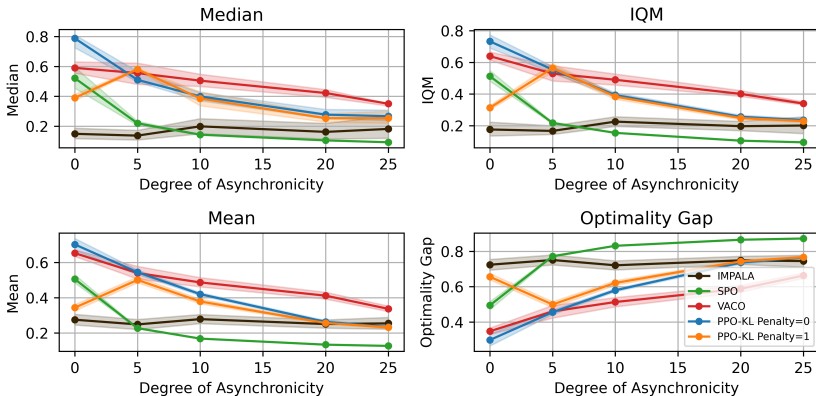

Figure 3: With more backward policy lag (higher degree of asynchrony) VACO achieves better performance on the aggregate metrics across various MuJoCo tasks. Higher Median, IQM, and Mean values and lower Optimality Gap imply better performance. The scores are computed over 100M steps across 10 independent random seeds and the shades represent 95% confidence intervals of the metrics.

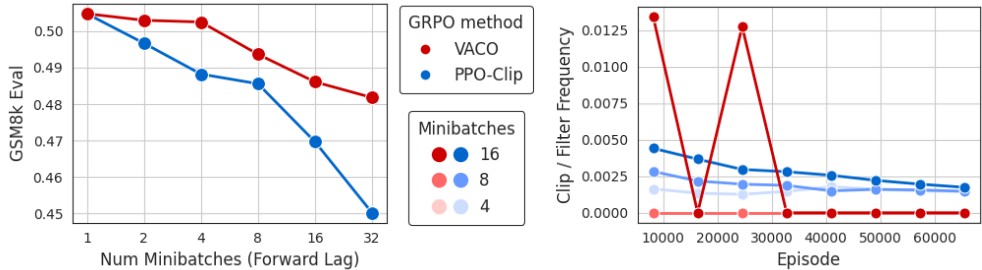

Figure 4: VACO improves over PPO-clipping for training LLMs to reason on GSM8k. **(Left)** Forward policy lag can improve training efficiency at the cost of eval performance. VACO maintains higher performance as lag increases. **(Right)** PPO-clip is always clipping, proportional to the forward lag. VACO filters more rarely, enabling learning from highly policy-lagged samples, but still maintains stability by filtering a larger part of the batch when activated.

use $\delta = 0.2$ as the constraint bound for VACO. Finally, in order to assess the effect of policy lag on the performance of the algorithms we use the MuJoCo environments that allow for high parallelization. As also noted in (Xie et al., 2025), since human scores are not available in MuJoCo locomotion tasks, we use the maximum and minimum score obtained from all the algorithms and normalize the scores as $\frac{\text{return-min}}{\text{max-min}}$.

Furthermore, as proposed by (Agarwal et al., 2021), we use stratified bootstrap confidence intervals and evaluate the confidence intervals of the algorithms to obtain various metrics. The aggregate metrics are calculated for each algorithm and for the runs with the fixed degree of asynchronous behavior (policy buffer capacity in Figure 1). The experimentation results are depicted in Figure 3. While higher degrees of asynchronicity affects all of the comparison algorithms, better robustness of VACO to off-policy data can be observed.

## 5.2 FORWARD POLICY LAG IN RLVR

Finally, we apply VACO to the modern setup of finetuning LLMs with RL to achieve reasoning using a verifiable reward (RLVR) (Lambert et al., 2025). A major challenge is that asynchronous RLVR (Noukhovitch et al., 2025) has been shown to be more efficient at the cost of increased policy lag which can degrade performance. Following state-of-the-art RLVR (Sheng et al., 2025) we generate $N$ minibatches of data using the same policy $\beta$ using our LLM inference engine, vllm Kwon et al.

(2023), label each answer with reward 1 (correct) or 0 (incorrect) and then training for $N$ steps using our training library, transformers (Wolf et al., 2020). Our policy $\pi$ therefore starts on-policy for the first minibatch but ends up with forward policy lag: $N-1$ steps ahead of our data generating policy $\beta$ by the final minibatch. We mostly follow the setup of Noukhovitch et al. (2025) and train a Qwen 2.5 0.5B base model (Qwen et al., 2025) on Grade School Math (GSM8k; Cobbe et al., 2021). We demonstrate the flexibility of our method by applying it to the standard algorithm in the field, GRPO (Shao et al., 2024), which can be seen as a modification of PPO that estimates advantages using Monte-Carlo sampling instead of a value function. All experimental details are in Appendix C.2.

We run our GRPO baseline using best PPO-clipping setup from current literature (Yu et al., 2025). We run 3 seeds for each minibatch $N \in 1, 2, 4, 8, 16, 32$ as large scale runs often use $N = 16$ minibatches or more (Yu et al., 2025). In Figure 4, we confirm previous findings (Noukhovitch et al., 2025) that GSM8k eval performance degrades from fully on-policy $N = 1$ as forward policy lag increases. Swapping PPO-clipping for VACO, we find greatly improved robustness to high amounts of forward lag. To show the difference, we plot the frequency of PPO-clipping and VACO filtering per batch for $N \in \{4, 8, 16\}$ in Figure 4 right. PPO-clipping is constant and the frequency increases in proportion to forward lag. In contrast, VACO does not do any filtering at low forward lag ($N = 4, 8$). For higher lag, VACO filters more selectively but a larger percentage of the batch. It is well-known that clipping is a key component of RLVR (Yu et al., 2025) but these results suggest that it is too aggressive and many clipped ratios positively contribute to learning. VACO still maintains the stability of training through filtering while allowing more batches to fully contribute to learning.

## 6 CONCLUSION

On-policy RL is the dominant paradigm across many modern RL tasks. As compute budgets increase, efficient methods that leverage asynchronous training and off-policy data have become standard. But these give rise to the phenomenon of policy lag i.e. the mismatch between the learning policy and the behavior policy. We propose a novel theoretical perspective that divides this phenomenon into backward and forward policy lag. Backward policy lag results from the initial mismatch between the learning and the behavior policy. Forward policy lag is caused by the learning policy taking gradient updates while using the same data. Asynchronous RL is inherently prone to both of these issues and they have been shown to degrade performance, limiting the efficiency gains from asynchronous training. This work provides a novel algorithm, VACO, designed to be more robust to the policy lag based on two main ideas: realigning the advantage function with the initial learning policy and Total Variation Divergence-based data filtering. Our empirical results demonstrate stronger robustness to lag in carefully constructed setups across two important RL modalities: robotics and LLMs. We hope this work enables future endeavors to further push the limits of asynchronous and efficient training while maintaining performance.

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

## A  PSEUDOCODE

---

**Algorithm 1** VACO

---

1: **Initialize** Initialize Value and Policy networks, $V_\phi, \pi_\theta^1$, and TV threshold $\delta$, value loss, policy loss, and entropy coefficients $c_v, c_\pi, c_\mathcal{H}$
2: **for** $T = 1, ..., T_{max}$ **do**
3:     # Data Collection Using a Behavior Policy
4:     Collect trajectory $\mathcal{D} = \{\tau = (s_t, a_t, r_t)_{t=0}^{\tau_{\max}} \sim \beta_T\}$
5:     # Estimate Advantage Using Advantage Realignment
6:     Calculate the Ratio $\frac{\pi_\theta^T(a_t|s_t)}{\beta_T(a_t|s_t)}$
7:     Calculate V-trace target $v(s_t, a_t)$ and Advantage $A_{vtrace}(s_t, a_t)$ using Equation 13 and 14
8:     **for** each training epoch **do**
9:         # Compute Expected TV Divergence
10:         $\bar{D}_{TV}(\pi_\theta \| \beta_T) = \frac{1}{2N} \sum_{t=1}^N |\frac{\pi_\theta(a_t|s_t)}{\beta_T(a_t|s_t)} - 1|$
11:         **if** $\bar{D}_{TV} > \delta/2$ **then**
12:             **for** $t = 1, ..., N$ **do**
13:                 **if** $(A_{vtrace}(s_t, a_t) - c_\mathcal{H}) \times \text{sgn}(\pi_\theta(a_t|s_t) - \beta_T(a_t|s_t)) > 0$ **then**
14:                     Detach Gradient $\pi_\theta(a_t|s_t)$
15:         $L_{\pi_\theta} \leftarrow -\frac{1}{N} \sum_{t=0}^N [\frac{\pi_\theta(a_t|s_t)}{\beta_T(a_t|s_t)} (A_{vtrace}(s_t, a_t) - c_\mathcal{H} \log \pi_\theta(a_t|s_t))$
16:     # Compute Value Loss
17:     $L_v \leftarrow \frac{1}{2N} \sum_{t=1}^N [V_\phi(s_t) - v(s_t, a_t)]$
18:     $L \leftarrow c_\pi L_\pi + c_v L_v$
19:     $\theta \leftarrow \theta - \eta_\theta \nabla_\theta L, \; \phi \leftarrow \phi - \eta_\phi \nabla_\phi L$

---

## B  THEORETICAL DETAILS

### B.1  PROOF OF LEMMA 1

We use (Agarwal et al., 2019) to provide the proof:

$$J(\pi') - J(\pi) = \mathbb{E}_{\tau \sim \pi' | s_0 \sim \mu}[\sum_{t=0}^\infty \gamma^t r(s_t, a_t)] - J(\pi) \tag{21}$$

$$= \mathbb{E}_{\tau \sim \pi' | s_0 \sim \mu}[\sum_{t=0}^\infty \gamma^t (r(s_t, a_t) + V_\pi(s_t) - V_\pi(s_t))] - J(\pi) \tag{22}$$

$$= \mathbb{E}_{\tau \sim \pi' | s_0 \sim \mu}[\sum_{t=0}^\infty \gamma^t (r(s_t, a_t) + \gamma V_\pi(s_{t+1}) - V_\pi(s_t))] \tag{23}$$

$$= \mathbb{E}_{\tau \sim \pi' | s_0 \sim \mu}[\sum_{t=0}^\infty \gamma^t (r(s_t, a_t) + \gamma \mathbb{E}[V_\pi(s_{t+1})|s_t, a_t] - V_\pi(s_t))] \tag{24}$$

$$= \mathbb{E}_{\tau \sim \pi' | s_0 \sim \mu}[\sum_{t=0}^\infty \gamma^t (Q_\pi(s_t, a_t) - V_\pi(s_t))] \tag{25}$$

$$= \mathbb{E}_{\tau \sim \pi' | s_0 \sim \mu}[\sum_{t=0}^\infty \gamma^t A_\pi(s_t, a_t)] \tag{26}$$

$$= \frac{1}{1 - \gamma} \mathbb{E}_{s \sim d^{\pi'}, a \sim \pi'(s)}[A_\pi(s, a)] \tag{27}$$

### B.2 PROOF OF THEOREM 1

First, we evaluate the change of state distribution in Equation 1:

$$J(\pi') - J(\pi) = \frac{1}{1-\gamma} \mathbb{E}_{s \sim d^{\pi'}, a \sim \pi'(s)}[A_\pi(s,a)] \tag{28}$$

$$= \frac{1}{1-\gamma}(\mathbb{E}_{s \sim d^\pi, a \sim \pi(s)}[\frac{\pi'(a|s)}{\pi(a|s)}A_\pi(s,a)] + \int_{s \in \mathcal{S}}(d^{\pi'}(s) - d^\pi(s))\mathbb{E}_{a \sim \pi'(s)}[A_\pi(s,a)]) \tag{29}$$

Furthermore, by applying Hölder's inequality to the penalty term, for any $p, q \in [1, \infty] : 1/p + 1/q = 1$ we have:

$$-\|d^{\pi'} - d^\pi\|_p \|\mathbb{E}_{a \sim \pi'(s)}[A_\pi(s,a)]\|_q \leq \int_{s \in \mathcal{S}}(d^{\pi'}(s) - d^\pi(s))\mathbb{E}_{a \sim \pi'(s)}[A_\pi(s,a)] \leq \|d^{\pi'} - d^\pi\|_p \|\mathbb{E}_{a \sim \pi'(s)}[A_\pi(s,a)]\|_q \tag{30}$$

Furthermore, the state distribution upper bound provided in (Achiam et al., 2017a) can be used:

**Lemma 4** *(Achiam et al., 2017a) The total variation divergence between two discounted future state distributions is bounded by the expected divergence their respective policies:*

$$\|d^{\pi'} - d^\pi\|_1 \leq \frac{2\gamma}{1-\gamma}\mathbb{E}_{s \sim d^\pi}[D_{TV}(\pi'\|\pi)[s]] \tag{31}$$

We refer readers to (Achiam et al., 2017a) for the detailed proof of the divergence lemma.

Choosing $p = 1, q = \infty$, applying Lemma 4 to Equation 30, and by bounding the Equation 28, the result of the lemma can be obtained.

### B.3 PROOF OF LEMMA 2

Use Theorem 1 to obtain the performance difference between $\pi_T$ and $\beta$ and between $\pi$ and $\beta_T$:

$$J(\pi) - J(\beta_T) \geq \frac{1}{1-\gamma}\mathbb{E}_{s \sim d^{\beta_T}, a \sim \beta_T(s)}[\frac{\pi(a|s)}{\beta_T(a|s)}A_{\beta_T}(s,a) - \frac{2\gamma\epsilon^\pi}{1-\gamma}D_{TV}(\beta_T\|\pi)[s]] \tag{32}$$

$$J(\pi_T) - J(\beta_T) \leq \frac{1}{1-\gamma}\mathbb{E}_{s \sim d^{\beta_T}, a \sim \beta_T(s)}[\frac{\pi_T(a|s)}{\beta_T(a|s)}A_{\beta_T}(s,a) + \frac{2\gamma\epsilon^{\pi_T}}{1-\gamma}D_{TV}(\beta_T\|\pi_T)[s]] \tag{33}$$

By inverting the second inequality and adding the two inequalities together, the bound can be achieved.

### B.4 PROOF OF LEMMA 3

The proof mainly follows the same steps as Theorem 1. Starting from Equation 3, change the discounted future state distribution from $\pi$ to $\beta_T$:

$$J(\pi) - J(\pi_T) = \frac{1}{1-\gamma}\left(\mathbb{E}_{s \sim d^{\beta_T}, a \sim \beta_T(s)}[\frac{\pi(a|s)}{\beta_T(a|s)}A_{\pi_T}(s,a)] + \int_{s \in \mathcal{S}}(d^{\beta_T}(s) - d^\pi(s))\mathbb{E}_{a \sim \pi(s)}[A_{\pi_T}(s,a)]ds\right) \tag{34}$$

Applying Hölder's inequality to the penalty term, for any $p, q \in [1, \infty] : 1/p + 1/q = 1$ we have:

$$\int_{s \in \mathcal{S}}(d^{\beta_T}(s) - d^\pi(s))\mathbb{E}_{a \sim \pi(s)}[A_{\pi_T}(s,a)]ds \leq \|d^{\beta_T} - d^\pi\|_p \|\mathbb{E}_{a \sim \pi(s)}[A_{\pi_T}(s,a)]\|_q \tag{35}$$

Choosing $p = 1, q = \infty$ and applying Lemma 4 provides the result.

## B.5 Theoretical Analysis of the Advantage Realignment

To analyze the approximation accuracy of the V-trace operator, (Espeholt et al., 2018) provides the following theorem:

**Theorem 2** *(Espeholt et al., 2018) Define the V-trace operator as:*

$$\mathcal{R}V(s) = V_\beta(s) + \mathbb{E}_{\tau \sim \beta}[\sum_{t=0}^{\infty} \gamma^t (c_0 \ldots c_{t-1}) \rho_t (r_t + \gamma V_\beta(s_{t+1}) - V_\beta(s_t)) | s_0 = s] \tag{36}$$

*Assume there exists $\alpha \in (0, 1]$ such that $\mathbb{E}_{a \sim \beta(s)}[\rho_0] \geq \alpha$ and let $\bar{\rho} \geq \bar{c}$ and the policy $\pi_{\bar{\rho}}$ as:*

$$\pi_{\bar{\rho}}(a|s) := \frac{\min(\bar{\rho}\beta(a|s), \pi(a|s))}{\sum_{b \in \mathcal{A}} \min(\bar{\rho}\beta(b|s), \pi(b|s))} \tag{37}$$

*Then $\mathcal{R}$ has a unique fixed point for $V_{\pi_{\bar{\rho}}}$ and is $\eta$-contraction mapping in sup-norm defined as:*

$$\eta = \gamma^{-1} - (\gamma^{-1} - 1)\mathbb{E}_{\tau \sim \beta}[\sum_{t \geq 0} \gamma^t (\prod_{i=0}^{t-1} c_i)\rho_{t-1}] \leq 1 - (1 - \gamma)\alpha < 1 \tag{38}$$

For completeness, we provide the proof of this theorem.

Notice that $\mathcal{R}$ can be rewritten as:

$$\mathcal{R}V(s) = (1 - \mathbb{E}_{a_0 \sim \beta(s)}[\rho_0])V_\beta(s) + \mathbb{E}_{\tau \sim \beta}\left[\sum_{t \geq 0} \gamma^t \Big(\prod_{i=0}^{t-1} c_i\Big)\Big(\rho_t r_t + \gamma[\rho_t - c_t \rho_{t+1}]V_\beta(s_{t+1})\Big)\right]. \tag{39}$$

Hence, we have:

$$\mathcal{R}V_1(s) - \mathcal{R}V_2(s) = (1 - \mathbb{E}_{a_0 \sim \beta(s)}[\rho_0])\big[V_\beta^1(s) - V_\beta^2(s)\big]$$

$$+ \mathbb{E}_{\tau \sim \beta}\left[\sum_{t \geq 0} \gamma^{t+1}\Big(\prod_{i=0}^{t-1} c_i\Big)[\rho_t - c_t \rho_{t+1}]\big[V_\beta^1(s_{t+1}) - V_\beta^2(s_{t+1})\big]\right] \tag{40}$$

$$= \mathbb{E}_{\tau \sim \beta}\left[\sum_{t \geq 0} \gamma^t\Big(\prod_{i=0}^{t-2} c_i\Big)[\rho_{t-1} - c_{t-1}\rho_t]\big[V_\beta^1(s_t) - V_\beta^2(s_t)\big]\right] \tag{41}$$

The authors define the notation $c_{-1} = \rho_{-1} = 1$ and $\prod_{s=0}^{t-2} c_s = 1$ for $t = 0$ and 1.

Furthermore, since it was assumed that $\rho \geq c$ and $\mathbb{E}_{\tau \sim \beta}[\rho_t] \leq \mathbb{E}_{\tau \sim \beta}[\frac{\pi(a_t|s_t)}{\beta(a_t|s_t)}]$, denoting $\alpha_t = \rho_{t-1} - c_{t-1}\rho_t$ we have:

$$\mathbb{E}_{\tau \sim \beta}\alpha_t = \mathbb{E}\big[\rho_{t-1} - c_{t-1}\rho_t\big] \geq \mathbb{E}_{\tau \sim \beta}\big[c_{t-1}(1 - \rho_t)\big] \geq 0 \tag{42}$$

Hence, the difference between the value functions $\mathcal{R}V_1(s) - \mathcal{R}V_2(s)$ is weighted linear combination of between the value functions in the trajectory $V_\beta^1(s_t) - V_\beta^2(s_t)$ with non-negative coefficients of

the sum:

$$\sum_{t \geq 0} \gamma^t \mathbb{E}_{\tau \sim \beta} \left[ \sum_{t \geq 0} \left( \prod_{i=0}^{t-2} c_i \right) [\rho_{t-1} - c_{t-1} \rho_t] \right] \tag{43}$$

$$= \sum_{t \geq 0} \gamma^t \mathbb{E}_{\tau \sim \beta} \left[ \left( \prod_{i=0}^{t-2} c_i \right) \rho_{t-1} \right] - \sum_{t \geq 0} \gamma^t \mathbb{E}_{\tau \sim \beta} \left[ \left( \prod_{i=0}^{t-2} c_i \right) \rho_t \right] \tag{44}$$

$$= \sum_{t \geq 0} \gamma^t \mathbb{E}_{\tau \sim \beta} \left[ \left( \prod_{i=0}^{t-2} c_i \right) \rho_{t-1} \right] - \frac{1}{\gamma} \left( \sum_{t \geq 0} \gamma^t \mathbb{E}_{\tau \sim \beta} \left[ \left( \prod_{i=0}^{t-2} c_i \right) \rho_{t-1} \right] - 1 \right) \tag{45}$$

$$= \gamma^{-1} - (\gamma^{-1} - 1) \sum_{t \geq 0} \gamma^t \mathbb{E}_{\tau \sim \beta} \left[ \left( \prod_{i=0}^{t-2} c_i \right) \rho_{t-1} \right] \tag{46}$$

$$\leq 1 - (1 - \gamma) \mathbb{E}_{a \sim \beta(s)} \rho_0 \tag{47}$$

$$\leq 1 - (1 - \gamma) \alpha \tag{48}$$

$$< 1 \tag{49}$$

Hence, it can be resulted that $\|\mathcal{R}V_1(s) - \mathcal{R}V_1(s)\| \leq \eta \|V_\beta^1(s) - V_\beta^2(s)\|_\infty$ with $1 - (1 - \gamma)\alpha \leq 1$ which makes $\mathcal{R}$ a contraction mapping.

Finally, to show that $\mathcal{R}$ is the fixed point of $\pi_{\bar{\rho}}$ we have:

$$\mathbb{E}_{\tau \sim \beta} [\rho_t (r(s_t, a_t) + \gamma V_{\pi_{\bar{\rho}}}(s_{t+1}) - V_{\pi_{\bar{\rho}}}(s_t) | s_t] \tag{50}$$

$$= \sum_{a \in \mathcal{A}} \beta(a|s_t) \min(\bar{\rho}, \frac{\pi(a|s_t)}{\beta(a|s_t)}) [r(s_t, a) + \gamma \sum_{s' \in \mathcal{S}} \mathcal{P}(s'|s_t, a) V_{\pi_{\bar{\rho}}}(s') - V_{\pi_{\bar{\rho}}}(s_t)] \tag{51}$$

$$= \sum_{a \in \mathcal{A}} \pi_{\bar{\rho}}(a|s_t) \left[ r(s_t, a) + \gamma \sum_{s' \in \mathcal{S}} \mathcal{P}(s'|s_t, a) V_{\pi_{\bar{\rho}}}(s') - V_{\pi_{\bar{\rho}}}(s_t) \right] \sum_{b \in \mathcal{A}} \min(\bar{\rho}\beta(b|s_t), \pi(b|s_t))$$
$$\tag{52}$$

$$= 0 \tag{53}$$

A zero Bellman error implies that $V_{\pi_{\bar{\rho}}}$ is the unique fixed point of the V-trace operator $\mathcal{R}$.

### B.6    ON THE USE OF TV DIVERGENCE AS A MEASURE OF POLICY LAG VS KL DIVERGENCE

Kullback–Leibler (KL) divergence is one of the most common measures of statistical distance between distribution. In this section, the benefits that the use of TV divergence as a measure of policy lag provides compared to KL divergence will be discussed. As discussed in Section 4.2, at iteration $T$, the performance difference between the two policy $\pi$ and $\pi_T$ when the trajectories are generated using a behaviour policy $\beta_T$ can be lower bounded using Equation 11.

Furthermore, as noted in Achiam et al. (2017a), using the Pinsker's inequality ($D_{TV}(p\|q) \leq \sqrt{D_{KL}(p\|q)/2}$) and Jensen's inequality, the expected TV divergence can be upper bounded as:

$$\mathbb{E}_{s \sim d^{\beta_T}} [D_{TV}(\beta_T \| \pi)[s]] \leq \sqrt{\mathbb{E}_{s \sim d^{\beta_T}} [D_{KL}(\beta_T \| \pi)[s]]/2} \tag{54}$$

By combining Equation 11 and Equation we have (a similar lower bound can also be derived from Equation 9):

$$J(\pi) - J(\pi_T) \geq \frac{1}{1 - \gamma} \mathbb{E}_{s \sim d^{\beta_T}, a \sim \beta_T(s)} \left[ \frac{\pi(a|s)}{\beta_T(a|s)} A_{\pi_T}(s, a) - \frac{2\gamma\epsilon^\pi}{1 - \gamma} D_{TV}(\beta_T \| \pi)[s] \right] \tag{55}$$

$$\geq \frac{1}{1 - \gamma} \mathbb{E}_{s \sim d^{\beta_T}, a \sim \beta_T(s)} \left[ \frac{\pi(a|s)}{\beta_T(a|s)} A_{\pi_T}(s, a) \right] - \frac{2\gamma\epsilon^\pi}{(1 - \gamma)^2} \sqrt{\frac{1}{2} \mathbb{E}_{s \sim d^{\beta_T}} [D_{KL}(\beta_T \| \pi)[s]]}$$

Table 1: Important Hyperparameters of PPO, SPO, IMPALA, and VACO

| Hyperparameter | Value |
| --- | --- |
| Clip Ratio/TV Threshold | 0.2 |
| Learning Rate | 3e-4 |
| Learning Rate Annealing | True |
| Num Envs | 500 |
| Num Steps | 1000 |
| Discount Factor | 0.99 |
| Num Minibatches | 32 |
| Num Epochs | 10 |
| Max Grad Norm | 0.5 |
| $\bar{\rho}$ | 1 |
| $\bar{c}$ | 1 |

For that reason, it is now a standard technique to use KL-regularized objective functions to optimize the policy Schulman et al. (2017; 2015a). However, we first note that the KL regularization is theoretically valid and guarantees policy improvement when $\sqrt{\mathbb{E}[D_{KL}(\beta_T \| \pi)[s]]} \geq 1$ whereas the thresholds that we often choose to constrain the KL divergence are usually less than one.

More importantly, KL divergence can pose more important optimization challenges. Consider Theorem 3 of Wang et al. (2020):

**Theorem 3** *Wang et al. (2020) Assume for discrete action space tasks with $|\mathcal{A}| > 3$, the policy parameterized by $\pi_\theta(s_t) = p_t \in \mathbb{R}^{+|\mathcal{A}|}$ with $\sum_{j=1}^{|\mathcal{A}|} p_t^{(i)} = 1$, and for continuous action space tasks with the policy parameterized as $\pi_\theta(a|s_t) = \mathcal{N}(a|\mu_t, \sigma_t)$. By defining the viable parameter space $\Theta = \{\theta | 1 - \delta \leq \frac{\pi_\theta(s,a)}{\beta_T(s,a)} \leq 1 + \delta \ \forall s, a \in \mathcal{S} \times \mathcal{A}\}$, then, $\max_{\theta \in \Theta} D_{KL}(\pi_\theta \| \beta_T)[s_t] = +\infty$ for both discrete and continuous action space tasks.*

Therefore, it is possible to have policies within the viable parameter space that impose infinite KL divergence. Hence, by framing the constrained policy optimization using $D_{KL} \leq \delta$, many viable policies would be dismissed. In contrast, it is easy to notice that all of the policies within the same parameter space $\Theta$, have $\max_{\theta \in \Theta} D_{TV}(\pi_\theta \| \beta_T)[s_t] \leq \delta/2$. Therefore, TV divergence provides a tighter lower bound for policy optimization and can capture a wider range of policies during the optimization process.

## C EXPERIMENTAL SETUP DETAILS

### C.1 SIMULATED ASYNCHRONOUS SETUP ALGORITHM HYPERPARAMETERS

We use the default hyperparameters proposed by Huang et al. (2022) to train the RL algorithms in the simulated asynchronous setup. Some of the important hyperparameters are discussed in Table 1. For the hyperparameter values used for the computation of the advantage realignment, we follow the hyperparameter choices of Espeholt et al. (2018).

### C.2 GSM8K RLVR SETUP AND HYPERPARAMETERS

We train on GSM8k mostly following the setup of Noukhovitch et al. (2025) but changing the model to the modern Qwen 2.5 0.5B base model (Qwen et al., 2025). We train on the $\approx$8000 examples of GSM8k (Cobbe et al., 2021) and evaluate on the $\approx$1300 examples of the eval set. Our hyperparameters mainly follow Yao et al. (2025) and we generally reproduce their $N = 1$ minibatch results, we note all hyperparameters in Table 2. To compare against the strongest possible baseline, we follow Yu et al. (2025) in using their separate low and high clipping values for GRPO.

As there is no backwards lag, we set the advantage realignment ratio to be 1. We note that an alternative approach would still use advantage realignment term, even in fully on-policy RLVR. As noted

by Yao et al. (2025), the logprobs of the data under the generation engine, vLLM, are different from those under the training library, transformers. For this reason, we could still implement advantage realignment with $\beta = \pi_{vllm}$, where it can be seen as an alternative formulation that solves the same issue as Truncated Importance Sampling Yao et al. (TIS; 2025). But for our particular scenario of both training and generation in BF16, Yao et al. (2025) note that the realignment term would be always very close to 1 and makes no empirical difference.

Table 2: Hyperparameters of GRPO, PPO-Clip and VACO for GSM8k

| Hyperparameter | Value |
|---|---|
| PPO-Clip Lower Ratio | 0.2 |
| PPO-Clip Higher Ratio | 0.272 |
| TV Threshold | 0.05 |
| Learning Rate | 1e-6 |
| Num Prompts Per Minibatch | 32 |
| Num Completions Per Prompt | 8 |
| Total Episodes | 65536 |
| Num Steps | 256 |
| Num PPO Epochs | 1 |
| Prompt Length | 512 |
| Completion Length | 512 |
| Temperature (Train) | 1.0 |
| Top-p (Train) | 1.0 |
| Temperature (Eval) | 0. |
| Top-p (Eval) 0.95 |  |

## D EXPERIMENT DETAILS

### D.1 ALGORITHM SAMPLE EFFICIENCY COMPARISON

To study the sample efficiency of VACO compared with other algorithms, Figure 5 shows the IQM value of the algorithms aggregated across the five environments versus the number of environment steps. Furthermore, the IQM values of the area under the curve of the normalized return plots demonstrates that VACO generally provides better sample efficiency across varying levels of asynchronicity. Figure 6 also showcases the return of the algorithms across all the environments and the degrees of asynchronicity. Additionally, Figure 7 presents the final returns of the algorithms in each environment.

### D.2 ABLATION ON THE VALUES OF $\delta$ THRESHOLD

We perform ablation analysis on the effect of the threshold $\delta$ on the performance of VACO. To this end, Figure 8 showcases the performance of VACO across a variety of values during the training process. The results indicate that VACO is relatively robust to more aggressive values for $\delta$. This point can be better observed in Figure 9. Hence, the constrained policy optimization approach of VACO is generally robust to the policy collapse in higher degrees of backward policy lag with more aggressive threshold values.

### D.3 ABLATION ON THE VALUES OF $\bar{\rho}$ THRESHOLD

We perform experiments on the effect of $\bar{\rho}$ on the performance of VACO. As discussed in Section 4.2 and Section B.5, $\bar{\rho}$ controls the fixed point of the V-trace target. Hence, the value can have important effects on the proposed algorithms. As shown in Figure 10 and Figure 11, our results confirm the original IMPALA paper Espeholt et al. (2018) regarding the overall better performance of IMPALA with $\bar{\rho} = 1$ compared with higher values.

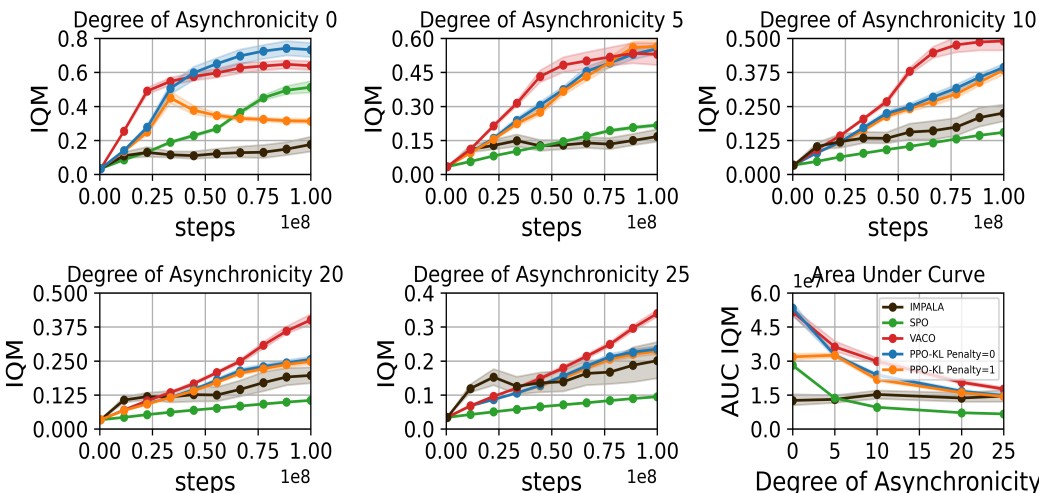

Figure 5: IQM values of the comparison algorithms in the simulated asynchronous setups during the training process. The scores are computed over 100M steps across 10 independent random seeds and the shades represent the 95% confidence interval.**(bottom right)** IQM values of the Area under the curve of the normalized return plots. Higher values imply better sample efficiency during the training process.

### D.4 EFFECT OF THE TV DIVERGENCE-BASED FILTERING

Figure 12 compares the average TV divergence of the final policy in each training in VACO and PPO with and without the KL penalty. We observe that while the value the TV divergence for PPO is lower, predicting its value from the clipping ratio would be a challenging task. On the other hand, the results for VACO indicates that the proposed algorithm maintains the same level of TV divergence. Hence, this behavior would allow the practitioners to reliably modify the constraint value to maintain a reasonably conservative TV divergence value while striving for optimality.

### D.5 EFFECT OF THE ADVANTAGE REALIGNMENT

In order to study the effect of Advantage Realignment in VACO we also ran the algorithm with and without the realignment process. The results are depicted in Figure 14 and Figure 13. An important effect that we observed during the training process was the fact that as more asynchronous data is added during the training process, the algorithm would perform better if the v-trace values are calculated using the most recent version of the value function. In fact, in some cases the algorithm did not perform well if the most recent version was not used.

## E USE OF LARGE LANGUAGE MODELS (LLMS)

In this work, LLM has mainly been for polishing the text and extracting grammatical errors. Furthermore, LLM was used in the process of writing and experimentation of the codebase and for creating the plots used in the paper.

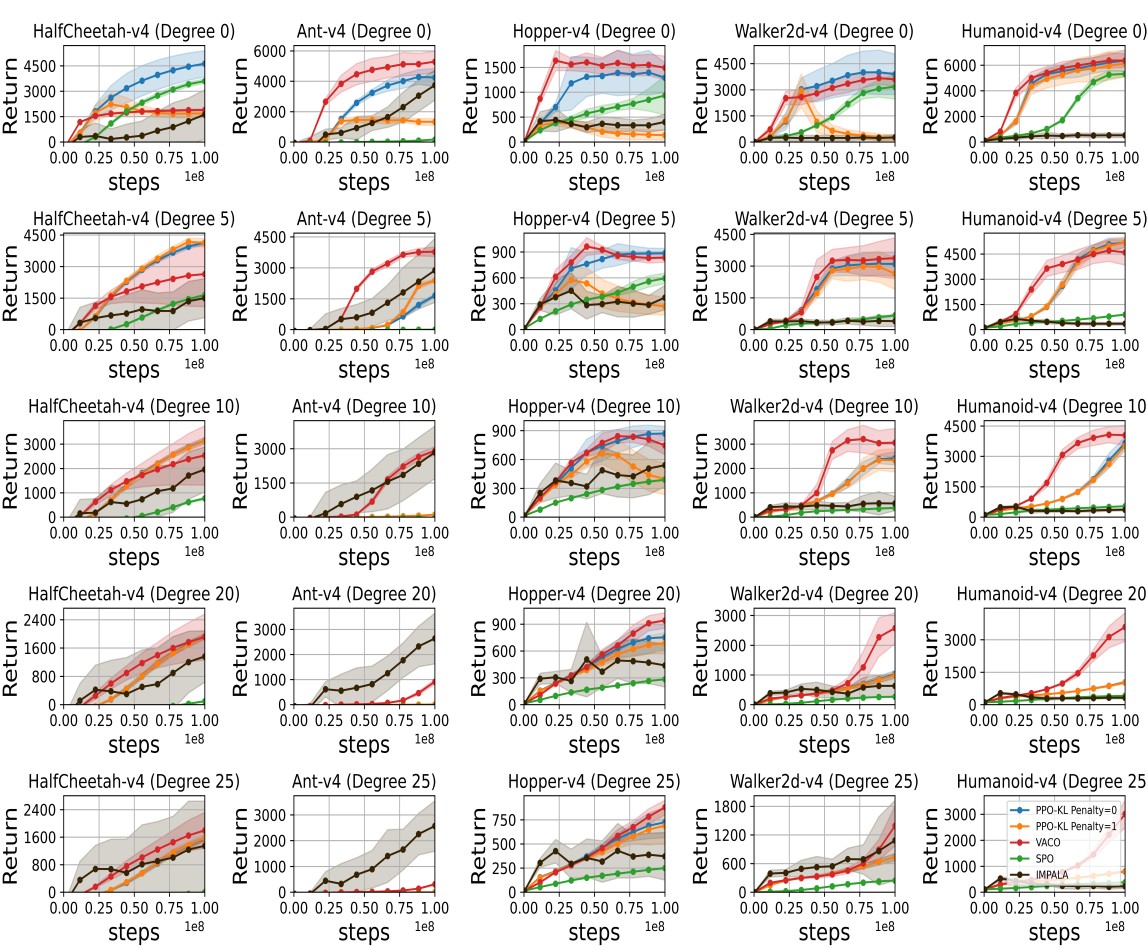

Figure 6: Return plot of the comparison algorithms in five environments across five degrees of asynchronicity. The plot represent the average return of the policy over 10 random seeds and the shades showcase the standard deviation of the returns.

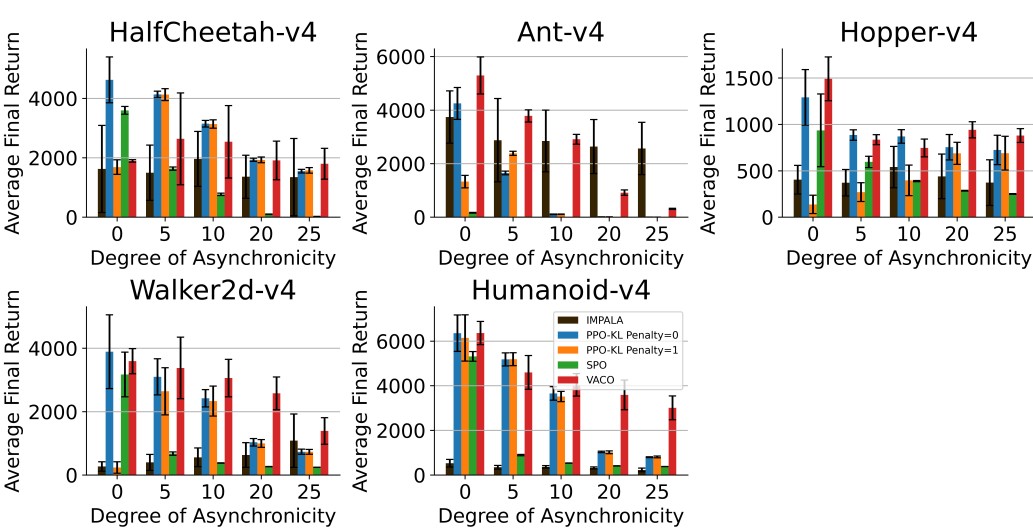

Figure 7: Average final returns of the comparison algorithms in the simulated asynchronous setup across various degrees of asynchronicity. The plot represent the average final return of the policy over 10 random seeds.

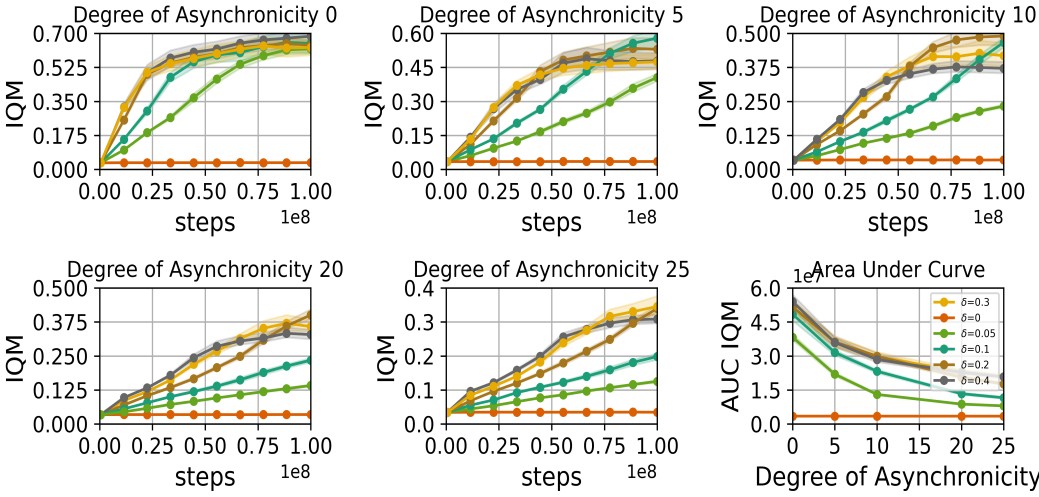

Figure 8: IQM values of VACO with various values of $\delta$ for TV divergence threshold in the simulated asynchronous setups during the training process. The scores are computed over 100M steps across 10 independent random seeds and the shades represent the 95% confidence interval. **(bottom right)** IQM values of the Area under the curve of the normalized return plots. Higher values imply better sample efficiency during the training process.

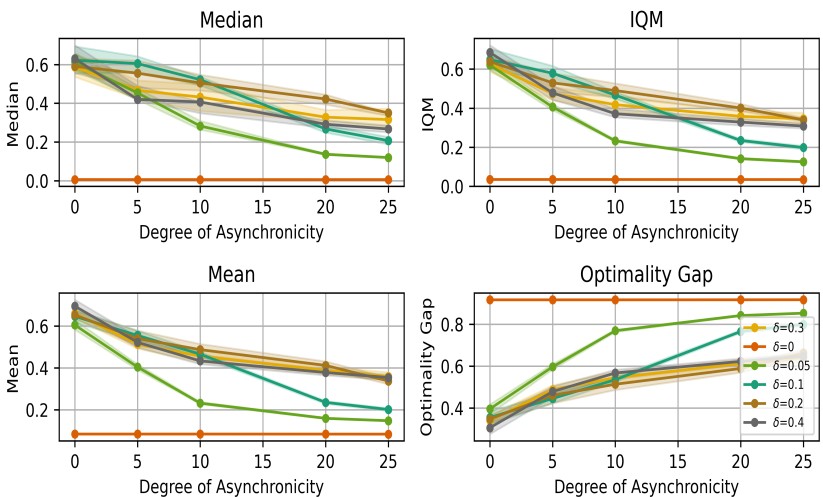

Figure 9: Ablation analysis on the performance of VACO with various value of the TV divergence threshold $\delta$. Higher Median, IQM, and Mean values and lower Optimality Gap imply better performance. The scores are computed over 100M steps across 10 independent random seeds and the shades represent 95% confidence intervals of the metrics.

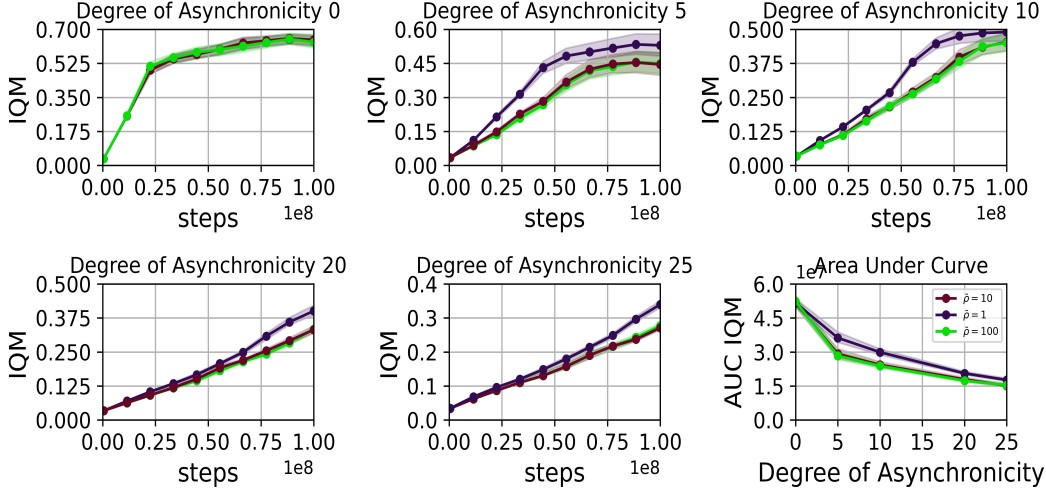

Figure 10: IQM values of VACO with various values of $\rho$ for advantage realignment in the simulated asynchronous setups during the training process. The scores are computed over 100M steps across 10 independent random seeds and the shades represent the 95% confidence interval. **(bottom right)** IQM values of the Area under the curve of the normalized return plots. Higher values imply better sample efficiency during the training process.

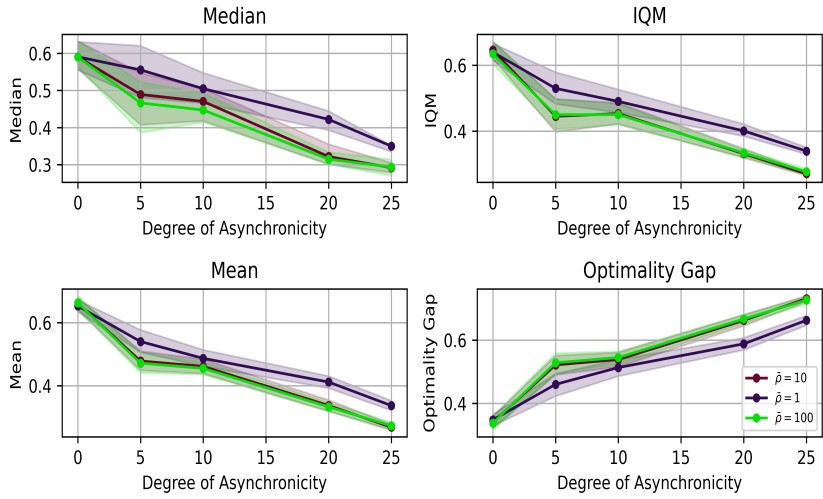

Figure 11: Ablation analysis on the performance of VACO with various value of $\rho$ for advantage realignment. Higher Median, IQM, and Mean values and lower Optimality Gap imply better performance. The scores are computed over 100M steps across 10 independent random seeds and the shades represent 95% confidence intervals of the metrics.

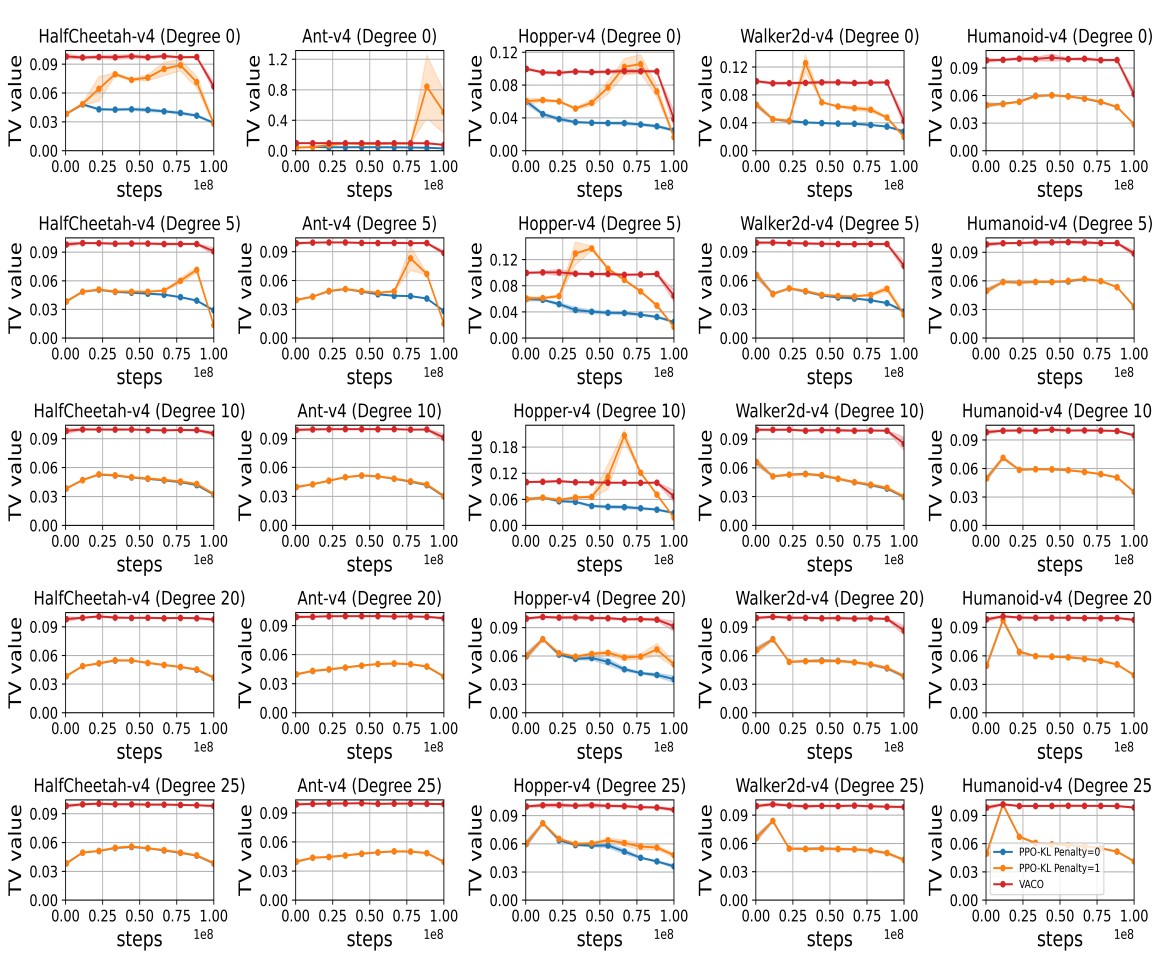

Figure 12: The average Total Variation Divergence between the final policy of the training phase and the behavior policy distribution for VACO and PPO in the simulated asynchronous RL training setup. We can see that VACO with the constraint of $1/2\mathbb{E}[|\pi/\beta - 1|] \leq \delta/2$ where $\delta = 0.2$, consistently maintains the same level of TV divergence across all environments and degrees of asynchronicity. The plot represent the average estimate of the TV value over 10 random seeds and the shades showcase the standard deviation.

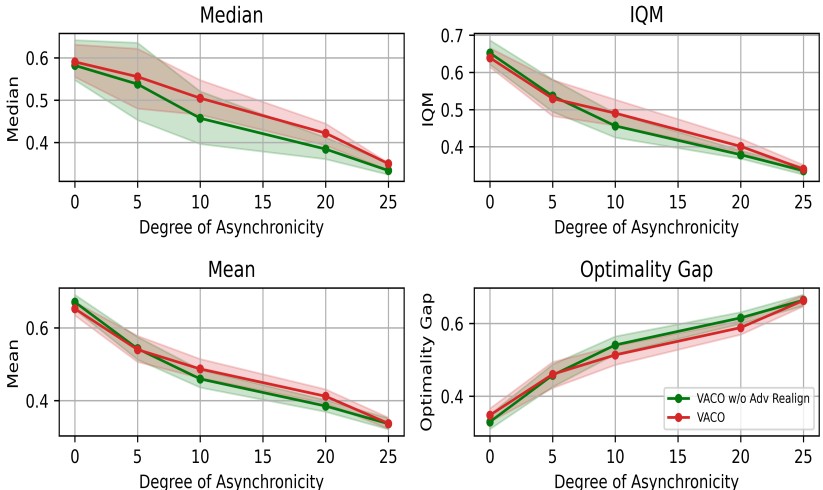

Figure 13: The performance of VACO with and without the advantage realignment process. We can see that on average, return realignment offers better robustness to off-policy data. The scores are computed over 100M steps across 10 independent random seeds and the shades represent the 95% confidence interval.

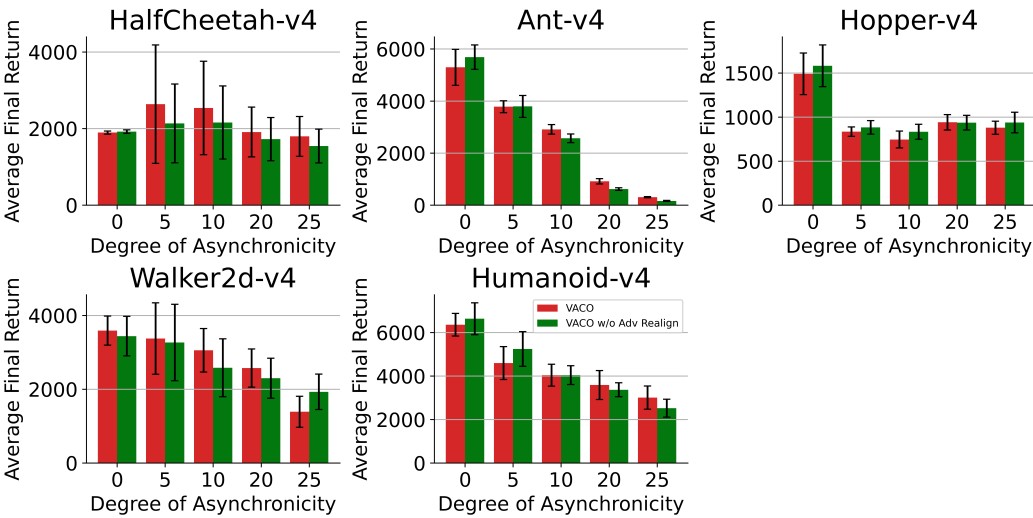

Figure 14: The performance of VACO with and without Advantage Realignment. VACO with the Realignment process achieves better overall performance with varying degrees of asynchronicity. The plot represent the average final return of the policy over 10 random seeds.

