# OpenReview forum: "Align and Filter: Improving Performance in Asynchronous On-Policy RL"
_ICLR.cc/2026/Conference — Submitted to ICLR 2026_

### Official Review · Reviewer_H8Np · 2025-10-27

**Soundness:** 4
**Presentation:** 2
**Contribution:** 4
**Rating:** 4
**Confidence:** 3

**Summary:**

The paper focuses on the policy lag issue in asynchronous on-policy reinforcement learning. The policy lag refers to the mismatch between the behavior policy used to collect the data and the evolution of the target policy while being updated. The paper analyzes the policy lag, noting that it is composed of two more fundamental components: the backward lag (due to an initial difference between the target policy and the asynchronously collected data), and the forward lag, due to the policy updates that further move the policy away from the collected data. On-policy methods like TRPO and PPO use a trust-region method designed to contain the forward policy lag. However, in PPO, the clipping surrogate objective does not effectively contain the policy lag.

The paper develops a novel update rule, combining two powerful ideas: one is to construct a total-variation gradient update that targets both components of the policy lag, while precisely filtering "bad gradients", and 2) better aligns the advantage estimation (which is used in the gradient updates) by using the v-trace target (from IMPALA) targeting the backward lag.

The novel algorithm (VACO: (Total Variation-based Advantage-aligned Constrained policy Optimization) can be used combined with other algorithmic architectures.

For this reason, the authors investigate the use of their updates alongside PPO on MuJoCo tasks and alongside GRPO on Grade School Math using an LLM architecture. The results obtained with 4 and 3 seeds, respectively, show improved performance compared to classic baselines.

**Strengths:**

This paper addresses a significant problem in asynchronous reinforcement learning. The forward aspect of the policy lag is also common in synchronous on-policy reinforcement learning (where consecutive policy updates actually create a distribution mismatch that is often overlooked in the on-policy literature).

The idea of using a total-variation filtering to prevent policy degradation and a "re-alignment" of the advantage estimation is sound.

The use of the performance difference lemma to show the backward and forward components of the policy lag is also sound, connecting the authors' intuition to an objective, mathematical understanding. The gradient in equation 15 is, to the best of my knowledge, novel, and the filtering mechanism presented in eq. 16 is sound.

Overall, the paper addresses a significant problem and proposes a sound, interesting analysis and solution.

**Weaknesses:**

The paper, in my opinion, contains several flaws that undermine the strengths highlighted above.

Clarity
---------

I find several aspects of the paper obscure. The most important thing is the mathematical notation. The policies $\pi, \pi_T, \beta_T$ have never been properly introduced. The description in Lemma 2 is not sufficient. What does "the iteration T" refer to? Collection iteration or policy improvement iteration? And why would the behavior policy change during the improvement iteration? And what is the difference between $\pi$ and $\pi_T$. This notation, used in the paper to present both the analysis and the results, must be clarified. Otherwise, it obscures the analysis and understanding of the algorithm. Figure 1 is not clear to me, as I am unsure what the authors want to convey. Figure 2, IMPALA vs Advantage Realignment, remains also obscure to me. I think I understand what is the difference beween the advantage realignment proposed by the authors and the one in IMPALA, but the figure itself, is not adding any information that I am able to understand.

The decision to use "advantage realignment" only at the beginning of the optimization seems poor or not well justified to me, especially given the author's objective of reducing the policy lag's impact on the policy update. IMPALA continuously re-aestimated the advantage exactly to target the "forward lag". I do not find convincing the authors's motivation that "this avoidance of repeated calculations makes their algorithm significantly more computationally efficient": what makes policy-gradient algorithms is computing gradients, but the V-trace simply corrects the GAE by using clipped importance sampling ratios. I would find much more credible to avoid repeating V-trace estimates in case they would increase too much the estimation variance (known problem for these kind of importance-sampling estimators). This point, if left unclarified, undermines the soundness of the proposed algorithm.

The authors claim in the introduction that the total variation captures both forward and backward lag (which I agree), but later state that the TV divergence targets the forward lag only (053--074).

Empirical Evaluations
-----------------------------

The paper's contribution revolves around two ideas: 1) using TV filtering and 2) using advantage re-alignment. I am unsure why the authors would not check the _independent_ contributions of these two tools. The empirical evaluation is solely focused on testing the whole recipe, leaving unquestioned the contributions of the two components.
In particular, 1) the intuition of TV-filtering depicted in Figure 2, where the filtering mechanism should remove gradients pointing outside the trust region, is not supported empirically. 2) The intuition that the advantage realignment improves the backward lag effect is also not supported empirically.

__Using four seeds for the MuJoCo tasks is absolutely not enough__. It is not clear on which data-aggregation level the median, quantile, and mean statistics refer: on four data points (one for each seed), or on n_environment x n_seed, or on n_episodes x n_environments x n_seeds?
The authors do not explain what the optimality gap is, and if it is simply 1 - normalized_score, it would be redundant information given the rest of Figure 3.
It is not clear to me why the authors use PPO-KL (the explanation in 410-411 is not sufficient to me), and why they do not show PPO-Clip.

The paper does not present any hyperparameter sensitivity.

Summary of Weakness Points
-----------------------------------------

1) Some fundamental parts of the paper have been poorly introduced.
2) It is not clear whether the TV filtering targets only the forward or both components of the policy lag.
3) It is not clear why the authors use the V-trace only at the beginning of the optimization, and not continuously: I find the computational efficiency argument unconvincing
4) The two main contributions of the paper have not been tested independently.
5) The number of seeds used in the paper is not sufficient to guarantee statistical significance. This can be understandable for the LLM experiment, but not for the MuJoCo environment.
6) It is not clear why the authors do not compare to PPO-Clip in the experimental section.
7) No hyperparameter sensitivity has been conducted (i.e., what is the influence of $\delta$).

**Questions:**

Suggestions/Questions
-------------------------------

1) Clarify all points I wrote(/questioned) above.
2) The sentence in 297--299 is unclear.
3) Balance the parentheses in equation 10 using \left and \right.
4) Figure 1 looks out of place, or is not well explained, and it is presented on page 2, while only referenced on page 8.
5) I would encourage the authors to present a bit better the whole asynchronous setup, contextualizing and clarifying the mathematical notation. This can be done in the background section.
6) The authors claim to have performed 100M steps in the MuJoCo environment. Is this a sum of all experiments/algorithms, or is it intended per single run? 100M per single run would seem excessive.

---

> ### Author Response · Authors · 2025-11-25
>
> We thank the reviewer for providing us with feedback. We try to address all of the concerns and questions raised. We are glad that the reviewer found our paper to contain “sound, interesting analysis and solution”. __We encourage the reviewer to read the revised manuscript, where all changes are highlighted in red.__
>
> > What does "the iteration T" refer to? Collection iteration or policy improvement iteration?
> And why would the behavior policy change during the improvement iteration?
>
> Iteration T is used to indicate one full phase of trajectory sampling (with behavior policy $\beta_T$) and policy optimization with multiple epochs (starting from the policy $\pi_T$). The behaviour policy is fixed during the policy optimization process. That is the reason why we used $\pi$ to indicate the most recent policy. We added the discussion in Section 3.2 and line 255 to clarify this point.
>
> > Figure 1 is not clear to me, as I am unsure what the authors want to convey. Figure 2, IMPALA vs Advantage Realignment, remains also obscure to me. I think I understand what is the difference beween the advantage realignment proposed by the authors and the one in IMPALA, but the figure itself, is not adding any information that I am able to understand.
>
> Figure 1 aims to differentiate between the real-world asynchronous setups and our proposed simulated setup. We used this figure to better convey the experimental setting we use to report the results. Furthermore, the advantage realignment vs IMPALA figure aims to showcase the main architectural difference between the two for the more general audience.
>
> > The decision to use "advantage realignment" only at the beginning of the optimization seems poor or not well justified to me, especially given the author's objective of reducing the policy lag's impact on the policy update. IMPALA continuously re-aestimated the advantage exactly to target the "forward lag". I do not find convincing the authors's motivation that "this avoidance of repeated calculations makes their algorithm significantly more computationally efficient": what makes policy-gradient algorithms is computing gradients, but the V-trace simply corrects the GAE by using clipped importance sampling ratios. I would find much more credible to avoid repeating V-trace estimates in case they would increase too much the estimation variance (known problem for these kind of importance-sampling estimators). This point, if left unclarified, undermines the soundness of the proposed algorithm.
>
> The main motivation for the use of advantage realignment only at the beginning mainly comes from the off-policy performance difference in Equation (9). The equation uses the advantage realignment to estimate the advantage of $\pi_T$ and remove the backward policy lag. In addition to the lower computational load that was discussed as an additional benefit compared to IMPALA, as the reviewer correctly mentions, VACO has lower sensitivity to advantage estimation inaccuracy. This allows VACO to have lower variance than IMPALA. Intuitively, VACO aims to offer both the data reusability in PPO and the robustness to asynchronous data in IMPALA. We added this discussion in Section 4.2.1 and to also better observe the performance difference between VACO and IMPALA, we implemented IMPALA on top of the CleanRL codebase and added the performance of IMPALA to the comparisons in figures 3,5,6,7. Note, however, that to the best of our knowledge there are no high-performing standard implementations of IMPALA in continuous action spaces. Our results indeed indicate that IMPALA generally has a high variance across various environments. In contrast, VACO offers more stable performance improvements.
>
> > The authors claim in the introduction that the total variation captures both forward and backward lag (which I agree), but later state that the TV divergence targets the forward lag only (053--074).
>
> We first claim that “Total Variation (TV)  divergence provides an effective tool for quantifying both sources of policy lag.” Here it is claimed that the TV divergence allows us to quantify both sources of policy lag.
>
> However, by observing the off-policy performance difference in Equation (9), it can be observed that we were able to remove the backward policy through the use of advantage realignment. Hence, our claim that “To control the forward policy lag that can arise from this process, we then use TV divergence to filter data points in each minibatch that would amplify the effect of forward policy lag.” aims to explain that our TV filtering method addresses the problem of forward policy lag in Equation (9).
> To avoid confusion, we explicitly added the description about the purpose of advantage realignment to address backward policy lag in line 53.

---

> > ### Author Response · Authors · 2025-11-25
> >
> > > The paper's contribution revolves around two ideas: 1) using TV filtering and 2) using advantage re-alignment. I am unsure why the authors would not check the independent contributions of these two tools.
> >
> > First, we would like to point out that we studied the effect of advantage realignment in Appendix D.5 and figures 13 and 14. Furthermore, we argue that the contribution of the advantage realignment cannot be independently studied without the TV filtering as the main policy optimization method (or some other optimization method) to control the effect of forward policy lag. However, we added an ablation analysis on the different values of delta as the TV threshold in Appendix D.2 to provide better insights into the effect delta on the performance of VACO. VACO shows robustness to policy collapse in higher degrees of backward policy lag with more aggressive threshold values (figures 8 and 9) while more conservative values can hinder its performance.
> >
> > > The empirical evaluation is solely focused on testing the whole recipe, leaving unquestioned the contributions of the two components. In particular, 1) the intuition of TV-filtering depicted in Figure 2, where the filtering mechanism should remove gradients pointing outside the trust region, is not supported empirically.
> >
> > We want to point out that we included the values of TV divergence during the training process in Appendix D.4. Figure 12 empirically showcases that VACO consistently maintains the same level of TV value during the training process.
> >
> > > 2) The intuition that the advantage realignment improves the backward lag effect is also not supported empirically.
> >
> > We perform an ablation analysis on the effect of advantage realignment in Appendix D.5 and show that, in figures 13 and 14, VACO achieves better performance with the advantage realignment as the degree of asynchronicity increases.
> >
> > > Using four seeds for the MuJoCo tasks is absolutely not enough.
> >
> > We updated and modified the figures 3,5 and figures 6 to 14 to showcase the performance of the algorithms and settings across 10 random seeds. Our results confirm the previous insights with better statistical significance. Specifically, we refer the reviewer to Figures 5 and 6 that showcases that VACO achieves high performance with a relatively low variance across 10 random seeds.
> >
> > > It is not clear on which data-aggregation level the median, quantile, and mean statistics refer: on four data points (one for each seed), or on n_environment x n_seed, or on n_episodes x n_environments x n_seeds?
> >
> > After each sampling-training iteration, we roll out the new policy in the environment and record the returns achieved by the policy. Hence, the statistics are calculated by averaging the achieved returns of the final trained policy for each seed. Therefore, each seed results in one data point. We reflected this experimental detail in the paper in Section 5.1.
> >
> > > The authors do not explain what the optimality gap is, and if it is simply 1 - normalized_score, it would be redundant information given the rest of Figure 3.
> >
> > We followed the approach proposed in [1] and the discussion in the caption of Figure 9 and section 4.3 in their paper. As noted there, “we recommend using the optimality gap: the amount by which the algorithm fails to meet a minimum score of γ = 1.0.”
> >
> > > It is not clear to me why the authors use PPO-KL (the explanation in 410-411 is not sufficient to me), and why they do not show PPO-Clip.
> >
> > There might have been a misunderstanding regarding this point. We did evaluate the vanilla version of PPO-Clip and named it in the figures as “PPO-KL Penalty=0” since PPO-KL has been considered the standard naming for when using the clipping mechanism along with the KL regularization. We updated section 5.1 to avoid the confusion.
> >
> > > The paper does not present any hyperparameter sensitivity.
> >
> > To address the reviewer’s concern, we added ablation results on the effect of the TV threshold and truncation value $\bar \rho$ in the appendices D.2 and D.3 as the two main hyperparameters in the algorithm. With respect to the TV threshold, VACO shows robustness to policy collapse in higher degrees of backward policy lag with more aggressive threshold values (figures 8 and 9) while more conservative values can hinder its performance. Moreover, we include ablation analysis on the truncation value $\bar \rho$ (figures 8 and 9) which confirms the insights provided in IMPALA regarding this value.
> >
> > >1. Clarify all points I wrote(/questioned) above.
> > >2. The sentence in 297--299 is unclear.
> > >3. Balance the parentheses in equation 10 using \left and \right.
> >
> > We would like to thank the reviewers for these suggestions. We fixed and refined these points in the paper.

---

> > > ### Author Response · Authors · 2025-11-25
> > >
> > > > 4. Figure 1 looks out of place, or is not well explained, and it is presented on page 2, while only referenced on page 8.
> > > > 5. I would encourage the authors to present a bit better the whole asynchronous setup, contextualizing and clarifying the mathematical notation. This can be done in the background section
> > >
> > > We added section 3.2 to better describe our notation in the context of asynchronous RL setup and added the motivation for using our simulated mixture of policies setup.
> > >
> > > > 6. The authors claim to have performed 100M steps in the MuJoCo environment. Is this a sum of all experiments/algorithms, or is it intended per single run? 100M per single run would seem excessive.
> > >
> > > For each environment, we highly parallelized 500 environments and rolled them out for 1000 steps in each iteration. Hence, the 100M steps in the environment is the total number of interactions of each algorithm with each environment. Our aim for choosing this setting was to evaluate various algorithms’ performance with the existence of extreme policy lag to show VACO’s robustness to different levels of policy lag over other algorithms.

---

### Official Review · Reviewer_75uv · 2025-10-31

**Soundness:** 3
**Presentation:** 3
**Contribution:** 3
**Rating:** 4
**Confidence:** 4

**Summary:**

This paper tackles policy lag in on-policy RL, and categorizes lag into backward lag and forward lag. This paper derives performance bounds using TVD and then propose VACO which combines advantage realignment and TV-based filtering. Experiments on MuJoCo and LLM reasoning tasks show the effectiveness of VACO.

**Strengths:**

1. Policy lag is a real bottleneck in scaling on-policy RL and LLM fine-tuning. The paper’s framing of forward vs backward lag is conceptually clear and useful.
2. Demonstrating both robotics and LLM reasoning experiments is good.

**Weaknesses:**

1. The filtering rule (Eq. 16) is heuristic; there is no formal proof that it guarantees constraint satisfaction or unbiased gradient estimates.
2. The use of TV divergence in continuous action spaces is only approximated by sample-based density ratios (Eq. 5).
3. 4 seeds are statistically insufficient for bootstrap confidence intervals, the CIs may not be meaningful.
4. Here is my major concern. It’s quite surprising that VACO needs 100 million steps to show improvements on MuJoCo, these are relatively simple continuous-control tasks where PPO usually reaches strong performance within a few million steps. That makes VACO’s sample efficiency look questionable. It seems the method only starts to outperform after very long training, possibly because the filtering and advantage realignment slow down learning. Also, the paper only shows final results, without any training curves, so it’s hard to tell whether VACO is truly more robust or just learning more slowly over time. If the authors can provide a clear explanation for why VACO requires such an extensive training horizon and clarify its sample efficiency compared to PPO, I would be willing to raise my score.

**Questions:**

1. The theory uses TV divergence, but empirical implementations rely on a sampled absolute ratio. Does this sample-based surrogate preserve the same monotonic improvement guarantees, especially in continuous space? Further, given the Pinsker's inequality, why not use the KL-divergence for your theory part? As I find in page 8, you refer to Pinsker'sinequality to minimize policy lag for PPO.

2. Claiming “no extra hyperparameters” is misleading. $\delta$ although is fixed to 0.2, but it seems to be a effective parameter. Also, how sensitive is VACO to $\delta$? Does the filtering frequency oscillate, causing instability across updates?

3. Is the ratio computed per dimension, per sample, or approximated by log-prob differences?

---

> ### Author Response · Authors · 2025-11-25
>
> We thank the reviewer for their feedback. We hope that we can address their concerns effectively. We are happy that the reviewer found our framing “conceptually clear and useful” and found our “robotics and LLM reasoning experiments good”. __We encourage the reviewer to read the revised manuscript, where all changes are highlighted in red.__
>
> > 1. The filtering rule (Eq. 16) is heuristic; there is no formal proof that it guarantees constraint satisfaction or unbiased gradient estimates.
>
> It is worth noting that the clipping mechanism in PPO and many of the more recent methods in on-policy RL are based on heuristics for which formal guarantees are challenging. While it is true that we have not provided a formal proof of the effectiveness of the filtering rule, we have shown its effectiveness in our experiments and figure 12 shows (appendix D.4) its capability in maintaining consistent levels of TV value.
>
> > 2. The use of TV divergence in continuous action spaces is only approximated by sample-based density ratios (Eq. 5).
>
> To the best of the authors’ knowledge there are no general case solutions that could be used for calculating the divergence between two Gaussian distributions. Moreover, while it is the case that in the continuous action case all of the policies are parameterized by a Gaussian distributions, in the asynchronous setup the action distribution of the behavior policy would no longer be Gaussian; rather, it is a mixture of Gaussians with a generally unknown mixing rule. Hence, by using the sampling formulation we can evaluate the TV divergence between complicated distributions without having any knowledge about their behaviors. We added Section 3.2 to discuss the specification of the behavior policy in our setup.
>
> > 3. 4 seeds are statistically insufficient for bootstrap confidence intervals, the CIs may not be meaningful.
>
> We updated and evaluated all of the algorithms and settings with 10 seeds to provide statistically more significant results in figures 3 and 5 to 14.  Our results confirm the previous insights with better statistical significance. We refer the reviewer to Figures 5 and 6 that showcases that VACO achieves high performance with a relatively low variance across 10 random seeds.
>
> > 4. Here is my major concern. It’s quite surprising that VACO needs 100 million steps to show improvements on MuJoCo, these are relatively simple continuous-control tasks where PPO usually reaches strong performance within a few million steps. That makes VACO’s sample efficiency look questionable. It seems the method only starts to outperform after very long training, possibly because the filtering and advantage realignment slow down learning. Also, the paper only shows final results, without any training curves, so it’s hard to tell whether VACO is truly more robust or just learning more slowly over time. If the authors can provide a clear explanation for why VACO requires such an extensive training horizon and clarify its sample efficiency compared to PPO, I would be willing to raise my score.
>
> We highly parallelized the environments to show the effect of asynchronous training similar to a real world distributed setup. In order to address the reviewer’s concern regarding the sample efficiency of VACO, we added figure 6 showing the returns of the algorithms per environment and per degree of asynchronicity versus training steps (see Appendix D.1).  In addition to that, in figure 5 we reported the IQM values of the normalized returns across the environments versus the steps for each degree of asynchronicity to better present overall sample efficiency of VACO during the training process for each degree of asynchronicity. Finally, the IQM values of the area under the curve of the normalized return plots versus the degree of asynchronicity was also added in figure 6 to provide a better understanding of the overall sample efficiency. Overall, at smaller degrees of asynchronicity, VACO performs on par with PPO. As the degree of asynchronicity increases, VACO shows better robustness both in terms of the sample efficiency throughout the training process and the final performance.
>
> > 1. The theory uses TV divergence, but empirical implementations rely on a sampled absolute ratio. Does this sample-based surrogate preserve the same monotonic improvement guarantees, especially in continuous space?
>
> Yes it does. This is mainly due to the fact that the expectation form of the TV divergence along with larger minibatch sizes, can provide a reasonable estimation of the TV divergence. The estimation accuracy of the sample-based expectation can be better observed from the TV divergence plots in figure 12 and the relatively low standard deviation during the training process.

---

> ### Author Response · Authors · 2025-11-25
>
> > Further, given the Pinsker's inequality, why not use the KL-divergence for your theory part? As I find in page 8, you refer to Pinsker'sinequality to minimize policy lag for PPO.
>
> Indeed, KL divergence is one of the most commonly-used measures of statistical distance in the literature. The main reason that we use the TV divergence for our theory was due to the fact that the measure of the policy lag through the TV divergence can provide a tight lower bound on the performance difference. However, the use of Pinsker’s inequality and further lower-bounding the performance difference can create a loose lower bound. We refer the reviewer to [1] and their Theorem 3 where they show that within the feasible parameter space $\Theta=\\{\theta|1-\delta\leq\frac{\pi_\theta(s,a)}{\beta_T(s,a)}\leq1+\delta~\forall s,a\in\mathcal{S}\times\mathcal{A}\\}$ for PPO, policies might suffer infinite KL divergence. This, however, is not the case with the TV divergence, and it provides a wider range of feasible policies. We added a detailed discussion on this comparison in the appendix B.6.
>
> [1] Wang, Yuhui, Hao He, and Xiaoyang Tan. "Truly proximal policy optimization." Uncertainty in artificial intelligence. PMLR, 2020.
>
> > 2. Claiming “no extra hyperparameters” is misleading. although delta is fixed to 0.2, but it seems to be a effective parameter.
>
> The claim “no extra hyperparameters” has been mainly made to indicate that our policy optimization method can effectively satisfy the TV constraint (specified through the value of $\delta$) without adding an extra layer of complexity for the choice of hyperparameters. In fact, as the ablation analysis of $\delta$ in appendix D.2 and figure 8 and 9 shows, we observed that the same set of parameters used for PPO would generally perform well with VACO which indicates that many of the engineering insights can be shared between VACO and PPO. It was for that reason that we chose $\delta$ to be of the same value as the clipping value in order to compare performances without specific hyperparameter tuning. To reflect this point, we rephrased this sentence as “We find that our filtering strategy is effective in maintaining a certain threshold on the value of the expected TV divergence without additional complexity for the choice of hyperparameters for constraint satisfaction” in line 56.
>
> > Also, how sensitive is VACO to delta? Does the filtering frequency oscillate, causing instability across updates?
>
> We fixed the threshold to match the hyperparameter values of PPO in the CleanRL codebase. To provide more insights into the effect of various parameters in VACO, we added ablation results on the effect of the TV threshold in appendix D.2. VACO shows robustness to policy collapse in higher degrees of backward policy lag with more aggressive threshold values (figures 8 and 9) while more conservative values can hinder its performance.
>
> > 3. Is the ratio computed per dimension, per sample, or approximated by log-prob differences?
>
> The ratios are computed for all of the data points in the minibatch sampled from the buffer. We use the ratios to both estimate the value of TV divergence and optimize the policy. We refer the reviewer to observe this detail in the Pseudocode section in Appendix A.

---

> ### Comment · Reviewer_75uv · 2025-11-25
> **Thanks for the detailed rebuttal!**
>
> Thank you very much for the detailed rebuttal and the effort put into revising the paper. I have reread the updated version, and most of my concerns have been addressed. Regarding points (1) and (2), my suggestion is to soften the claim of theoretical grounding and instead make the theoretical gap explicit, positioning the theory primarily as motivation. In addition, for the filtering mechanism, providing more explanation of the intuition behind it would further help readers understand its role. Overall, I will increase my score. Good luck with your submission.

---

> > ### Author Response · Authors · 2025-11-26
> >
> > We thank the reviewer for their engagement and kind words. We will update the manuscript based on the reviewer's suggestions during the next round of revisions. We would also like to kindly ask the reviewer if there is anything else we can address that would make the reviewer more strongly supportive of our work, please let us know.

---

### Official Review · Reviewer_rsdg · 2025-11-01

**Soundness:** 3
**Presentation:** 2
**Contribution:** 2
**Rating:** 2
**Confidence:** 3

**Summary:**

This paper studies policy lag in asynchronous on-policy RL. In particular, they paper defines two types of lag: backward lag (a mismatch between behavior and learner at the start of optimization), and forward lag (policy starts diverging from the data distribution after multiple updates on the same batch of data). To mitigate lag, the paper proposes VACO which has two core features: (1) advantage realignment, a V-trace-style off-policy estimate aligned to the learner, and (2) TV-divergence–based filtering which removes per-sample gradients predicted to increase TV beyond a threshold. Experiments show robustness benefits over PPO and SPO in MuJoCo tasks and in a simple RLVR tasks.

**Strengths:**

1. The paper theoretically characterizes lag. While it seems obvious to me that lag would interfere with policy optimization, it's very useful to show this mathematically.
2. Experiments on both RLVR for LLMs and more standard RL tasks (MuJoCo) is a nice plus. The "RL for LLMs" feels too distinct from the broader RL community, so I'm glad that people are writing papers that evaluate on both setups.

**Weaknesses:**

I lean to reject due to limited experiments and missing discussion on existing works for managing off-policy-ness during on-policy learning.

1. **On-policy learning with Off-policy data**. My understanding is that the policy lag problem is simply the problem of trying to perform on-policy updates with off-policy data. Several works have studied how to leverage off-policy data for on-policy updates, especially importance sampling methods like Queeney et al 2021, but the only baselines considered are PPO with different KL penalties and SPO. In the  Why not compare to importance sampling methods? Is there something about the asynchronous setup such that making existing off-policy correction methods impractical?

2. **Statistical Significance.** Results in Figure 3 are averaged over 4 seeds, which is too few to draw reliable conclusions. Prior work (Henderson et al., AAAI 2018) shows that even identical RL algorithms can appear statistically distinct when evaluated on a small number of trails (like n=5).

3. **Novelty.** To my knowledge, the formal analysis of lag is novel, though VACO itself is quite similar to TRPO’s TV-based bounds and IMPALA’s V-trace. The paper claims two differences: (1) single-shot advantage realignment (vs. IMPALA’s continuous realignment) and per-sample TV-aligned filtering (vs. PPO clipping). What can VACO achieve that a well-tuned KL-penalized PPO or SPO cannot?

**Questions:**

1. To preface this question, I think the decision to focus on on-policy methods in this paper is reasonable and justified. Nevertheless, in asynchronous settings with significant lag, wouldn't it make more sense to revert back to off-policy methods rather than try to learn on-policy with increasingly off-policy data?

---

> ### Author Response · Authors · 2025-11-25
>
> We thank the reviewer for their comments and questions and hope that our response can address their concerns. We are glad that the reviewer found our mathematical characterization of policy lag “very useful” and found our “experiments on both RLVR for LLM and more standard RL tasks a nice plus”. __We encourage the reviewer to read the revised manuscript, where all changes are highlighted in red.__
>
> > 1. __On-policy learning with Off-policy data.__ My understanding is that the policy lag problem is simply the problem of trying to perform on-policy updates with off-policy data. Several works have studied how to leverage off-policy data for on-policy updates, especially importance sampling methods like Queeney et al 2021, but the only baselines considered are PPO with different KL penalties and SPO. In the Why not compare to importance sampling methods? Is there something about the asynchronous setup such that making existing off-policy correction methods impractical?
>
> In the context of on-policy RL, most of the prior works (especially in the case of Queeney et al 2021) assume access to the previous policies or a certain replay buffer with the knowledge about the policies that have generated the transitions. However, in the asynchronous setups, and especially in the more modern distributed RL architectures, having the knowledge about the past policies requires undesirable, constraining engineering and considerations. Our aim is to frame and analyze the problem theoretically and provide a simple, general-purpose algorithm capable of providing robustness to policy lag with minimal assumptions about the setup. Moreover, to address the reviewer’s concern regarding the off-policy correction methods, we implemented IMPALA on top of the CleanRL codebase and added the performance of IMPALA to the comparisons. Note, however, that to the best of our knowledge there are no high-performing standard implementations of IMPALA in continuous action spaces. We find that VACO overall performs better than our version of IMPALA in Figures 3,5,and 6.
>
> > 2. __Statistical Significance.__ Results in Figure 3 are averaged over 4 seeds, which is too few to draw reliable conclusions. Prior work (Henderson et al., AAAI 2018) shows that even identical RL algorithms can appear statistically distinct when evaluated on a small number of trails (like n=5).
>
> We updated and evaluated all of the algorithms and settings with 10 seeds to provide statistically more significant results in figures 3 and 5 to 14. Our results confirm the previous insights with better statistical significance. Specifically, we refer the reviewer to Figures 5 and 6 that showcases that VACO achieves high performance with a relatively low variance across 10 random seeds.

---

> ### Author Response · Authors · 2025-11-25
>
> > 3. __Novelty.__ To my knowledge, the formal analysis of lag is novel, though VACO itself is quite similar to TRPO’s TV-based bounds and IMPALA’s V-trace. The paper claims two differences: (1) single-shot advantage realignment (vs. IMPALA’s continuous realignment) and per-sample TV-aligned filtering (vs. PPO clipping). What can VACO achieve that a well-tuned KL-penalized PPO or SPO cannot?
>
> Our results show SPO’s brittleness to the asynchronous setup, which might be due to it not being designed with the asynchronous setup in mind. In the case of KL-penalized PPO, while a well-tuned KL coefficient can offer a good performance, we argue that the process of finding this value adds another layer of complexity to getting the favorable performance.
> In contrast, we empirically showed that VACO is able to effectively satisfy the specified constraint $\delta$ on the TV divergence which effectively addresses the same challenge that the clipping mechanism and the KL penalty aim to address. Moreover, as discussed, the TV filtering method directly sets the magnitude of the forward policy lag allowing the practitioners to better control it. We refer the reviewer to the TV divergence plot in the appendix D.4  and figure 12 to observe the effectiveness of TV filtering in maintaining the same value of the TV value throughout the training process.
>
> In addition to the hyperparameter selection, one of the main reasons that we use the TV divergence for our theory is due to the fact that the measure of policy lag through the TV divergence can provide a tight lower bound on the performance difference. However, the use of Pinsker’s inequality and further lower-bounding the performance difference can create a loose lower bound. We refer the reviewer to [1] and their Theorem 3 where they show that within the feasible parameter space $\Theta=\\{\theta|1-\delta\leq\frac{\pi_\theta(s,a)}{\beta_T(s,a)}\leq1+\delta~\forall s,a\in\mathcal{S}\times\mathcal{A}\\}$ that PPO aims to optimize for, policies might suffer infinite KL divergence. This, however, is not the case with the TV divergence, and it provides a wider range of feasible policies. We added a detailed discussion on this comparison in the appendix B.6.
>
> [1] Wang, Yuhui, Hao He, and Xiaoyang Tan. "Truly proximal policy optimization." Uncertainty in artificial intelligence. PMLR, 2020.
>
> > To preface this question, I think the decision to focus on on-policy methods in this paper is reasonable and justified. Nevertheless, in asynchronous settings with significant lag, wouldn't it make more sense to revert back to off-policy methods rather than try to learn on-policy with increasingly off-policy data?
>
> Our aim for presenting the asynchronous experiments was to evaluate the performance of various algorithms with the existence of extreme policy lag to show VACO’s robustness. On-policy methods are very popular in many environments including for distributed training and in some cases (such as RL for LLMs) using off-policy methods may not be scalable. For those reasons, we argue that being able to address extreme policy lag in asynchronous setups can be valuable to the community.

---

### Official Review · Reviewer_MnL1 · 2025-11-01

**Soundness:** 3
**Presentation:** 3
**Contribution:** 3
**Rating:** 6
**Confidence:** 3

**Summary:**

The paper proposes an RL algorithm named Total Variation–based Advantage-aligned Constrained policy Optimization (VACO) to mitigate policy lag in distributed/asynchronous on-policy RL. It provides supporting theory and validates the approach on MuJoCo control and LLM reasoning tasks.

**Strengths:**

1. The paper offers solid, well-structured theoretical analysis that clearly supports the method.

2. The proposed approach targets a relevant problem in distributed/on-policy RL.

3. The experiments stay focused on the core question, with detailed comparisons provided in the appendix.

**Weaknesses:**

Limited sensitivity analysis of the TV threshold. The method fixes the throshold and does not ablate it. It’s unclear how training stability and performance depends on throshold ,

**Questions:**

Could the authors provide an ablation and guidance on choosing the throshold in practice?

---

> ### Author Response · Authors · 2025-11-25
>
> We thank the reviewer for their comments and we aim to address their question accordingly. We are happy that the reviewer found the paper to have “well-structured theoretical analysis” and that “the experiments focused on the core question”. __We encourage the reviewer to read the revised manuscript, where all changes are highlighted in red.__
>
>
> > Limited sensitivity analysis of the TV threshold. The method fixes the threshold and does not ablate it. It’s unclear how training stability and performance depends on threshold
>
> We fixed the threshold to match the hyperparameter values of PPO in the CleanRL codebase. To provide more insights into the effect of various parameters in VACO, we added ablation results on the effect of the TV threshold (in Appendix D.2) and truncation value $\bar \rho$ in (Appendix D.3). With respect to the TV threshold, VACO shows robustness to the policy collapse in higher degrees of backward policy lag with more aggressive threshold values (figures 8 and 9) while more conservative values can hinder its performance. Moreover,  we added or updated the figures 3 and 6 to 14 to provide better presentation. We included sample efficiency plots (figures 5 and 6) to show VACO’s better performance throughout the training process and included the truncation value $\bar \rho$ ablation analysis (figures 8 and 9) which confirms the insights provided in IMPALA regarding this value.
>
>
> > Could the authors provide an ablation and guidance on choosing the threshold in practice?
>
> As the ablation analysis on the value of delta in appendix D.2 and figures 8 and 9 shows, we observed that the same set of parameters used for PPO would generally perform well with VACO which indicates that many of the engineering insights can be shared between VACO and PPO.

---

### Author Response · Authors · 2025-12-04

We would like to thank all the reviewers for their valuable comments and feedback and the ACs for their efforts.

We will provide a summary of the reviews and the revisions we made to the manuscript.

The reviewers generally found our problem important and the theoretical analysis clear and useful. They found our proposed solution novel and interesting, and they were interested in our experiments on both “RL for LLMs” and robotics tasks, which are generally two of the important use cases of on-policy RL in distributed setups.

To address reviewer concerns, we revised the manuscript (__the changes are highlighted in red in the manuscript__), and we summarize the important changes below.

>__1) Ablation/Sensitivity analysis:__ The reviewers were generally concerned that the paper provided limited ablation analysis regarding the performance of VACO. Reviewers __MnL1__, __75uv__, and __H8Np__ were concerned that “Limited sensitivity analysis of the TV threshold. The method fixes the threshold and does not ablate it.”, “how sensitive is VACO to delta?”, and “The paper does not present any hyperparameter sensitivity.”.

To address the concern of the reviewers we add multiple ablation analyses on different components of the algorithm. In Appendix D.2 we study the effect of different values of the threshold $\delta$ on the performance of VACO and observed that more conservative values might hinder the performance of VACO while higher threshold values would generally result in similar performance demonstrating the robustness of VACO to higher thresholds.

In addition to that we perform ablation analysis on the effect of the threshold value $\bar \rho$ on the performance of VACO in Appendix D.3 and confirmed the insights provided by IMPALA paper on $\bar \rho=1$ generally providing better performance across various levels of asynchronicity.

In Appendix D.4 we provide the value of TV divergence throughout the training process to showcase that within various asynchronicity settings and different environments VACO consistently maintains the same level of TV divergence during the training process which makes it a reliable tool at controlling the policy lag.

In Appendix D.5 we study the effect of Advantage Realignment on the performance of VACO. This is also in line with the reviewer __H8Np__ comments regarding “The intuition that the advantage realignment improves the backward lag effect is also not supported empirically”. We show that advantage realignment can improve the training performance across various levels of asynchronicity.

>__2) Comparison with Off-policy importance sampling correction methods:__ Reviewers __rsdg__ and __H8Np__ were concerned about the differences between VACO and IMPALA. __rsdg__ asked “In the Why not compare to importance sampling methods? Is there something about the asynchronous setup such that making existing off-policy correction methods impractical?” and __H8Np__ asked “The decision to use "advantage realignment" only at the beginning of the optimization seems poor or not well justified to me, especially given the author's objective of reducing the policy lag's impact on the policy update. IMPALA continuously re-estimated the advantage exactly to target the "forward lag".

While there are no high performance standard implementations of IMPALA in continuous action spaces, we implemented IMPALA (as one of the most important importance sampling correction methods) on top of the CleanRL codebase to compare it with the other algorithms. As noted in our discussion with __H8Np__ and our comparison results, IMPALA generally has a higher variance in performance which may be due its sensitivity to advantage estimation inaccuracies. On the other hand, VACO provides a more stable performance by controlling the forward policy lag through the use of the TV divergence rather than re-estimating the advantage function for every new policy. We added this discussion to Section 4.2.1 and updated the figures 3,5,6,7 to add the performance of IMPALA.

> __3) Statistical Significance:__ Reviewers __rsdg, 75uv,__ and __H8Np__ were concerned that using 4 seeds to compare the algorithm is too few to draw statistically significant conclusions.

To address this concern we updated and evaluated all of the algorithms and settings with 10 seeds to provide statistically more significant results in figures 3 and 5 to 14. Our results confirm the previous insights with better statistical significance. Specifically, Figures 5 and 6 showcase that VACO achieves high performance with a relatively low variance across 10 random seeds.

---

> ### Author Response · Authors · 2025-12-04
>
> > __4) Fundamental advantage to the use of TV filtering compared with PPO and SPO methods:__ The reviewer __rsdg__ asked the important question that “What can VACO achieve that a well-tuned KL-penalized PPO or SPO cannot?”.
>
> First we pointed out that our results show SPO’s brittleness to the asynchronous setup, which might be due to it not being designed with the asynchronous setup in mind. In the case of KL-penalized PPO, while a well-tuned KL coefficient can offer a good performance, we argue that the process of finding this value adds another layer of complexity to getting the favorable performance. In contrast, we empirically showed that VACO is able to effectively satisfy the specified constraint on the TV divergence which addresses the same challenge that the clipping mechanism and the KL penalty aim to address (again we refer the AC to Appendix D.4).
>
> In addition to that, also in line with the reviewer __75uv__ question “given the Pinsker's inequality, why not use the KL-divergence for your theory part?”, we provided a detailed comparison between the use of TV divergence and KL divergence and argued that within the feasible parameter space $\Theta=\\{\theta|1-\delta\leq\frac{\pi_\theta(s,a)}{\pi_T(s,a)}\leq1+\delta~\forall s,a\in\mathcal{S}\times\mathcal{A}\\}$, the TV divergence can capture all of the feasible policies whereas the policies in the same parameter space may suffer infinite KL divergence. Therefore, formulating the constraint using the KL divergence may disregard many feasible policies. Hence, the TV divergence can provide a wider range of feasible policies. We added a detailed discussion section regarding this comparison in Appendix B.6.
>
> > __5) Sample Efficiency:__ Reviewer __75uv__ had the important concern about why VACO needs 100M steps to show better performance compared to other algorithms. They asked “That makes VACO’s sample efficiency look questionable. It seems the method only starts to outperform after very long training, possibly because the filtering and advantage realignment slow down learning”.
>
> To provide better context regarding the sample efficiency of VACO compared to other algorithms we added the algorithm comparisons during the training process in Appendix D.1. In Figure 5 we provide the IQM value of the algorithms across different environments for each degree of asynchronicity. In addition to that, we also provide the Area Under the Curve of the normalized return plots for each degree of asynchronicity. Both sets of plots indicate that VACO not only achieves better final performance, but also provides better sample efficiency throughout the training process. In addition to that, in figure 6 we provide the return plot of the algorithms for each environment and degree of asynchronicity.
>
> > __6) Notation and Presentation:__ One of the important notational concerns was raised by reviewer H8Np. They asked “The notation, used in the paper to present both the analysis and the results, must be clarified.” and suggested “I would encourage the authors to present a bit better the whole asynchronous setup, contextualizing and clarifying the mathematical notation. This can be done in the background section.”
>
> To address the reviewer’s concern and provide a better context to the asynchronous setup, we add Section 3.2 which discusses the mathematical setup that the asynchronous setup follows and provides a clearer explanation for why we use the mixture of policies to simulate the asynchronous setup. We also introduced the obscure notations the reviewer was concerned about in that section.
>
> > __7) Effectivity of TV filtering:__ The reviewers __75uv__ and __H8Np__ were concerned that the TV filtering solution that we proposed is a heuristic and no theoretical guarantees are provided to support its effectiveness. __75uv__ raised “The filtering rule (Eq. 16) is heuristic; there is no formal proof that it guarantees constraint satisfaction or unbiased gradient estimates.” and __H8Np__ asked “the intuition of TV-filtering depicted in Figure 2, where the filtering mechanism should remove gradients pointing outside the trust region, is not supported empirically.”
>
> We first noted that the clipping mechanism in PPO and many of the more recent methods in on-policy RL are based on heuristics for which formal guarantees are challenging. While it is true that we have not provided a formal proof of the effectiveness of the filtering rule, we have shown its effectiveness in our experiments and figure 12 (appendix D.4) shows its performance in maintaining consistent levels of TV value which empirically demonstrates its capability in constraint satisfaction across all the environments and degrees of asynchronicity.

---

### Meta-Review · Area_Chair_AJVe · 2025-12-05

**Summary:**

This paper studies policy lag in asynchronous on-policy RL. In particular, the paper defines two types of lag: backward lag (a mismatch between behavior and learner at the start of optimization), and forward lag (policy starts diverging from the data distribution after multiple updates on the same batch of data). To mitigate lag, the paper proposes VACO, which has two core features: (1) advantage realignment, a V-trace-style off-policy estimate aligned to the learner, and (2) TV-divergence–based filtering, which removes per-sample gradients predicted to increase TV beyond a threshold. Experiments show robustness benefits over PPO and SPO in MuJoCo tasks and in a simple RLVR task.

Multiple reviewers have acknowledged the value of characterizing policy lag. But reviewers also pointed out a series of presentation and clarity issues, statistical significance during the evaluation, and the need for more comparisons/discussions with relevant methods. Some of these concerns have been addressed, while some still remain. I think this submission could benefit from another round of careful refinement, which can further strengthen the paper.

**Reviewer Concerns:**

Some concerns that I think are still not well-addressed:
- Novelty issues and lack of discussion on existing works for managing off-policy-ness during on-policy learning (Reviewer rsdg)
- Theory vs. practical implementation gap (Reviewer 75uv)
- Clarity of the paper (Reviewer 75uv, H8Np). Partly addressed during rebuttal, but could benefit more from further polishing.
- Seemingly poor sample efficiency and requirement of prolonged training (Reviewer 75uv). Only partly addressed during the rebuttal.

Some comments that I believe have been addressed or partly addressed during rebuttal, based on additional experimental results:
- Statistical significance of using 4 seeds (Reviewer rsdg, 75uv, H8Np). Addressed during rebuttal by increasing to 10 seeds.
- Insufficient fine-grained ablation (separate impact of using TV filtering and advantage re-alignment) (reviewer H8Np).

**Reviewer Scores:**

Reviewer 75uv mentioned he/she will increase the score.

Reviewer rsdg had not responded during the rebuttal. I think even if he were to increase the score, it might still fall on the negative end.

---

### Decision · Program_Chairs · 2026-01-26

Reject